

# Automated forward and adjoint modelling of viscoelastic deformation of the solid Earth

William Scott[1], Mark Hoggard[1], Thomas Duvernay[1], Sia Ghelichkhan[1,2], Angus Gibson[1], Dale Roberts[1], Stephan C. Kramer[3], and D. Rhodri Davies[1]

[1]Research School of Earth Sciences, The Australian National University, Canberra, ACT, Australia.
[2]Institute for Water Futures, The Australian National University, Canberra, ACT, Australia.
[3]Department of Earth Science and Engineering, Imperial College London, London, UK.

**Correspondence:** William Scott (william.scott1@anu.edu.au)

**Abstract.**

Robust models of viscoelastic Earth deformation under evolving surface loads underscore many problems in geodynamics and are particularly critical for paleoclimate and sea-level studies through their role in Glacial Isostatic Adjustment (GIA). A long-standing challenge in GIA research is to perform computationally efficient inversions for ice-loading histories and man-
tle structure using a physically realistic Earth model that incorporates three-dimensional viscosity variations and/or complex rheologies. For example, recent geodetic observations from melting ice sheets appear inconsistent with long-term sea-level records and have been used to argue for transient rheologies, generating debate in the literature and leaving large uncertainties in projections of future sea-level change. Here, we extend the applicability of G-ADOPT (a Firedrake-based finite element framework for geoscientific adjoint optimisation) to these problems. Our implementation solves the equations governing vis-
coelastic surface loading while naturally accommodating elastic compressibility, lateral viscosity variations, and non-Maxwell rheologies (including transience). We benchmark the approach against a suite of analytical and numerical test cases, demonstrating both accuracy and computational efficiency. Crucially, G-ADOPT enables automatic derivation of adjoint sensitivity kernels, allowing gradient-based optimisation strategies that are essential for high-dimensional inverse problems. Using synthetic Earth-like experiments, we illustrate its capability to reconstruct ice histories and recover mantle viscosity variations,
providing a roadmap towards data assimilation and uncertainty quantification in GIA modelling and sea-level projections.

## 1 Introduction

Modelling the solid Earth's viscoelastic response under evolving surface loads is a foundational problem in geodynamics that has a rich history spanning more than a century of research (e.g. Gilbert, 1886; Haskell, 1935; Wu and Peltier, 1982; Mitrovica, 1996). Surface loads of geological interest can be of both natural and anthropogenic origin and operate across a
wide range of spatial and temporal scales — from metres to thousands of kilometres, and from minutes to millions of years. Pertinent examples include coseismic deformation (e.g. Pollitz, 1996), ocean and body tides (e.g. Dehant et al., 1999; Lau et al., 2017; Matviichuk et al., 2020), reservoir impoundment (e.g. Chao et al., 2008), redistribution of surface and groundwater (e.g. Crittenden Jr, 1963; Crowley et al., 2006; Austermann et al., 2020), volcanic loading (e.g. Moore, 1970; Rundle, 1982), and





the formation and melting of ice sheets (e.g. Haskell, 1935; Cathles, 2016). Understanding these processes has required the
development of increasingly complex models of deformation within Earth's interior and at its surface.

     One of the most societally relevant applications of viscoelastic modelling is glacial isostatic adjustment (GIA), a process
that links the evolution of ice sheets to solid Earth deformation and sea-level change (Daly, 1925; Haskell, 1935; Farrell and
Clark, 1976; Peltier et al., 2015). Due to the combined effects of viscoelastic uplift and subsidence, gravitational changes, and
feedbacks on Earth's rotation, GIA introduces strong geographic variability into sea-level change (e.g. Mitrovica et al., 2011).
Attempts to constrain modern sea-level trends driven by climate change are further complicated by the fact that, due to the
mantle's high viscosity, Earth is still responding to the retreat of major ice sheets over Fennoscandia and North America since
the Last Glacial Maximum (LGM) around 20,000 years ago (e.g. Lambeck et al., 2014). All of these factors must be considered
when interpreting geodetic observations such as Global Navigation Satellite System (GNSS) and GRACE data (Church et al.,
2004; Tregoning et al., 2009; King et al., 2012; Ivins et al., 2013; Caron and Ivins, 2020) and a complete understanding of GIA
– past, present and future – is critical in generating robust projections of future sea level.

     Viscoelastic surface-loading phenomena can generally be considered as inverse problems. The rheological properties of
the solid Earth are poorly known, and observations of its deformation in response to various loads must therefore be used to
infer these properties. This situation is compounded in the case of GIA, since present-day observables such as crustal uplift
rates or geoid changes are jointly controlled by rheological properties and the history of ice-ocean loading — both of which
are poorly constrained. Progress in this field has relied on fitting combinations of ice history and Earth structure to diverse
datasets, including relative sea level (RSL) indicators (e.g. tide gauge records, fossil corals, paleo strand-lines), geomorphic
reconstructions of past ice extent (e.g. terminal morraines, cosmogenic exposure ages), GNSS surface motions, satellite gravity
data, and indirect constraints from seismology and other palaeoclimate archives (Lambeck et al., 2014; Peltier et al., 2015;
Gowan et al., 2021).

Most GIA models exploring these datasets assume a radially symmetric (1D) Earth and represent its viscoelastic response
with a Maxwell rheology (Wu and Peltier, 1982). However, we know from mantle convection studies that mantle rheology
deviates significantly from this axisymmetric assumption, being strongly dependent on temperature and strain-rate among
other factors (e.g. Moresi and Solomatov, 1998; Tackley, 2000), and resulting in lateral viscosity variations that span several
orders of magnitude. Sea-level records and geodetic observations confirm that such 3D viscosity variations exert a first-order
influence on GIA deformation patterns (Wu and van der Wal, 2003; Austermann et al., 2013; Nield et al., 2014; Hay et al., 2017;
Barletta et al., 2018; Powell et al., 2020; Austermann et al., 2021). This recognition has driven the development of 3D GIA
models using finite volume (Latychev et al., 2005), finite element (Wu, 2004; Zwinger et al., 2020; Weerdesteijn et al., 2023),
and spectral approaches (Martinec, 2000; Tanaka et al., 2011). At the same time, both laboratory deformation experiments and
seismic anisotropy observations indicate that non-Maxwell rheologies, including power-law creep, are likely to be important,
at least in the upper mantle (Karato and Wu, 1993). In these cases, the strain rate has a non-linear dependence on driving stress,
which has been explored in some more advanced GIA models (Wu, 1995; Gasperini et al., 2004; van der Wal et al., 2010).
Furthermore, both laboratory experiments and observations of post-seismic deformation suggest that Earth's rheology may also
depend on the frequency of loading (i.e., exhibit transient behaviour; Karato and Wu, 1993; Pollitz, 2003; Pollitz et al., 2006;





Faul and Jackson, 2015; Yamauchi and Takei, 2016; Lau, 2024). In regions such as West Antarctica and coastal Greenland,
recent observations of rapid GNSS-measured uplift rates, compared with longer-term post-LGM RSL curves, have sparked
renewed debate, with both transient rheology and 3D viscosity variations proposed as viable explanations (e.g. Nield et al.,
2014, 2018; Barletta et al., 2018; Adhikari et al., 2021; Lau et al., 2021; Pan et al., 2024; Weerdesteijn and Conrad, 2024).

While the importance of accounting for 3D Earth structure and potentially complex rheologies in GIA is now beyond doubt,
3D GIA models suffer from being computationally expensive. Conventional 'brute-force' inversion strategies, whereby model
sensitivities are assessed by running many forward simulations, are therefore impractical. Consequently, most 3D GIA models
are still run in a forward sense and rely on LGM ice histories independently derived from 1D inversions and 3D Earth structures
inferred from other datasets, such as seismic tomography. A transformative alternative to brute-force inversion is gradient-based
optimisation, which is enabled by adjoint methods. This technique allows efficient computation of model sensitivities – that is,
the gradient of a misfit function with respect to a large number of model parameters (or 'controls'). Unlike in a finite difference
approach, where cost scales with the number of control parameters (which is often tied to mesh resolution), the adjoint method
computes gradients at a cost comparable to a single linearised forward solve, independent of the number of control parameters
(Errico, 1997). This aspect makes it uniquely scalable and powerful for PDE-constrained optimisation and it is now widely
adopted in fields such as numerical weather prediction (Kalnay, 2003), oceanography (Forget et al., 2015), seismic imaging
(Tromp et al., 2005; Fichtner et al., 2006), and mantle convection (Ghelichkhan et al., 2021).

In geodynamics, the past decade has seen substantial theoretical progress in applying adjoint methods to viscoelastic loading
problems. In the case of GIA, their use was first demonstrated by Al-Attar and Tromp (2014), with extension to sea-level
modelling on a 1D Earth undertaken by Crawford et al. (2018) and on a 3D Earth by Lloyd et al. (2024). These developments
have revealed strong diagnostic power: adjoint sensitivity kernels clearly expose the limitations of using a 1D Earth and the first-
order impact of lateral viscosity variations on GIA observables. As such, they are increasingly recognised as a game-changing
tool for tackling the spatial complexity and high dimensionality of the GIA inverse problem. Furthermore, recent advances in
second-order adjoint techniques may offer a tantalising route to formal uncertainty quantification through efficient Hessian-
based inference (e.g. Yu et al., 2025). Nevertheless, there remain obstacles to the broader use of adjoint techniques within
the GIA modelling community. A central challenge lies in the complexity of deriving, implementing, and validating adjoint
models for large-scale, non-linear, time-dependent problems. Compounding this is the legacy of traditional GIA workflows,
which have typically relied on 1D Earth structures and relatively simple rheologies.

The shift toward 3D solid Earth structure, non-linear rheologies, and formal data assimilation places new demands on model
infrastructure, underscoring the need for automated, scalable, and efficient adjoint-capable frameworks. The automatic adjoint
framework provided by the Geoscientific Adjoint Optimisation Platform (G-ADOPT; Davies et al., 2022; Ghelichkhan et al.,
2024; Gibson et al., 2024) offers a compelling and timely solution. G-ADOPT combines three state-of-the-art software libraries:
(i) Firedrake, a flexible and automated system for solving partial differential equations using the finite element method (Ham
et al., 2023); (ii) Pyadjoint, which enables automatic derivation of discrete adjoints within Firedrake (Farrell et al., 2013;
Mitusch et al., 2019); and (iii) the Rapid Optimisation Library (ROL), a high-performance optimisation engine within Trilinos



(The ROL Project Team, 2022). Together, these components deliver a scalable, modular and performant modelling framework that can robustly handle complex forward models and compute their adjoints with theoretical optimal efficiency.

In this paper, we extend, benchmark and demonstrate the applicability of G-ADOPT for solving viscoelastic surface-loading problems. The framework accommodates arbitrary rheologies, including transient behaviour, and can operate in both Cartesian and spherical geometries. We verify its accuracy against analytical solutions (Section 3.1), benchmark it against published 3D Cartesian models under time-dependent loading scenarios (Section 3.2), and extend it to incorporate compressible elasticity, Burgers rheology (Section 3.3), and spherical domains with lateral viscosity variations (Section 3.4). Finally, we use a series

of twin experiments (Section 3.5) to showcase the framework's adjoint-based inversion capabilities, recovering unknown ice-loading histories and mantle structure in a 2D annulus. Although self-gravitation is not yet included, these developments represent a significant step towards a next-generation, data-assimilating framework for GIA and other surface-loading problems within G-ADOPT.

## 2   Method

In this section we present the non-dimensional equations governing the viscoelastic loading problem, consisting of the conservation laws of mass and momentum driven by an evolving surface load. The non-dimensional mass conservation equation is given by

$$\partial_t \rho + \nabla \cdot (\rho \boldsymbol{v}) = 0, \tag{1}$$

where $\boldsymbol{v}$ and $\rho$ denote the non-dimensional velocity and density, respectively. In non-dimensionalising the velocity, we have

used the scale $\frac{L}{\bar{\alpha}}$, where $L$ is a characteristic length scale and $\bar{\alpha}$ is a characteristic Maxwell time that describes the transition between dominantly elastic and viscous behaviour

$$\bar{\alpha} = \frac{\bar{\eta}}{\bar{\mu}}, \tag{2}$$

where $\bar{\eta}$ and $\bar{\mu}$ are characteristic dynamic viscosity and shear modulus values respectively (see Table 1).

    The density field is decomposed into a time-independent background component, $\rho_0$, and a small time-dependent perturbation, $\rho_1$, such that $\rho = \rho_0 + \rho_1$. Substituting this decomposition into Eq. (1) yields the linearised mass conservation equation

$$\partial_t \rho_1 + \nabla \cdot (\rho_0 \boldsymbol{v}) = 0, \tag{3}$$

where we have neglected the $\nabla \cdot (\rho_1 \boldsymbol{v})$ term, by assuming that perturbations in density are small compared with background values. Integrating this equation in time and using the relation $\partial_t \boldsymbol{u} = \boldsymbol{v}$, where $\boldsymbol{u}$ is the non-dimensional displacement vector,

we obtain the following expression for the density perturbation

$$\rho_1(t) = -\nabla \cdot (\rho_0 \boldsymbol{u}(t)) = -\rho_0 \nabla \cdot \boldsymbol{u}(t) - \boldsymbol{u}(t) \cdot \nabla \rho_0, \tag{4}$$





assuming a vanishing initial displacement, $\boldsymbol{u}(t=0)=\boldsymbol{0}$.

The full GIA equations include rotational and gravitational effects arising from changes in the surface load as water is redistributed between ice and ocean, which is governed by the sea-level equation (Kendall et al., 2005). Neglecting those
terms, alongside inertial terms, yields the following non-dimensional form of the momentum equation

$$\boldsymbol{0} = \nabla \cdot \boldsymbol{\sigma} - B_\mu \left( \rho_0 + \rho_1 \right) g \, \hat{\boldsymbol{e}}_k, \tag{5}$$

where $\boldsymbol{\sigma}$ is the non-dimensional stress tensor, $g$ is the non-dimensional gravity, and $B_\mu = \frac{\bar{\rho}\bar{g}L}{\bar{\mu}}$ is a non-dimensional number describing the ratio of buoyancy to elastic shear strength. Note that $\hat{\boldsymbol{e}}_k$ is aligned with either the $z$-axis or radial direction in Cartesian or spherical coordinates, respectively. We refer again to Table 1 for the characteristic non-dimensional scales.

To further simplify the momentum equation, we linearise perturbations in the stress field by expressing the total stress as a sum of a time-invariant hydrostatic background component and a perturbation: $\boldsymbol{\sigma} = \boldsymbol{\sigma}_0 + \boldsymbol{\sigma}_1^E$, where the superscript $E$ indicates an Eulerian perturbation. Since viscoelastic constitutive relations that link stress to strain (and their time derivatives) are fundamentally defined in a Lagrangian framework, we also need to transition from an Eulerian to a Lagrangian description of the stress field. To capture the material response in a Lagrangian framework, the perturbation must account for the change in
stress due to displacement of material through the background stress gradient. To first order, the incremental Lagrangian stress is $\boldsymbol{\sigma}_1^L = \boldsymbol{\sigma}_1^E + \boldsymbol{u} \cdot \nabla \boldsymbol{\sigma}_0$ (see Eq. 3.16 of Dahlen and Tromp, 1998). Substituting into Eq. (5), we obtain

$$\boldsymbol{0} = \nabla \cdot \left( \boldsymbol{\sigma}_0 + \boldsymbol{\sigma}_1^L - \boldsymbol{u} \cdot \nabla \boldsymbol{\sigma}_0 \right) - B_\mu \left( \rho_0 + \rho_1 \right) g \, \hat{\boldsymbol{e}}_k. \tag{6}$$

We assume that time-invariant components are in equilibrium, such that the background stress field satisfies a hydrostatic balance with the background density

$$\nabla \cdot \boldsymbol{\sigma}_0 = B_\mu \, \rho_0 \, g \, \hat{\boldsymbol{e}}_k. \tag{7}$$

Moreover, assuming that the background density $\rho_0$ varies only in the radial direction, the advection of hydrostatic pre-stress simplifies to

$$\boldsymbol{u} \cdot \nabla \boldsymbol{\sigma}_0 = B_\mu \, \rho_0 \, g \, u_k \, \boldsymbol{I}, \tag{8}$$

where $u_k$ is the component of non-dimensional displacement in the $\hat{\boldsymbol{e}}_k$ direction and $\boldsymbol{I}$ is the identity tensor. Substituting these
expressions leads to the final linearised form of the momentum equation

$$\boldsymbol{0} = \nabla \cdot \boldsymbol{\sigma}_1^L - B_\mu \nabla \left( \rho_0 g u_k \right) - B_\mu \rho_1 g \, \hat{\boldsymbol{e}}_k. \tag{9}$$

## 2.1 Constitutive Equation

We follow the internal variable formulation adopted by Al-Attar and Tromp (2014) and Crawford et al. (2017, 2018), in which viscoelastic constitutive equations are expressed in integral form and reformulated using so-called *internal variables*.
Conceptually, this approach consists of a set of elements with different shear relaxation timescales, arranged in parallel.





**Table 1.** Summary of notation and model parameters.

| Symbol | Description | Value |
|--------|-------------|-------|
| $\boldsymbol{\sigma}$ | Non-dimensional stress tensor | - |
| $\boldsymbol{\sigma_0}$ | Non-dimensional background stress tensor | - |
| $\boldsymbol{\sigma}_1^E$ | Non-dimensional incremental Eulerian stress tensor | - |
| $\boldsymbol{\sigma}_1^L$ | Non-dimensional incremental Lagrangian stress tensor | - |
| $\boldsymbol{m}$ | Non-dimensional internal variable | - |
| $\boldsymbol{d}$ | Non-dimensional deviatoric strain tensor | - |
| $\boldsymbol{e}$ | Non-dimensional strain tensor | - |
| $\boldsymbol{v}$ | Non-dimensional velocity | - |
| $\boldsymbol{u}$ | Non-dimensional displacement | - |
| $u_k$ | Non-dimensional displacement in direction of gravity | - |
| $\rho$ | Non-dimensional density | - |
| $\rho_0$ | Non-dimensional background density | - |
| $\rho_1$ | Non-dimensional Eulerian density perturbation | - |
| $\bar{\rho}$ | Characteristic density scale | $4500 \ \mathrm{kg/m^3}$ |
| $\rho_{\mathrm{load}}$ | Non-dimensional load density | - |
| $L$ | Characteristic length scale | $3 \times 10^6 \ \mathrm{m}$ |
| $\kappa$ | Non-dimensional bulk modulus | - |
| $\mu$ | Non-dimensional shear modulus | - |
| $\mu_0$ | Non-dimensional effective shear modulus | - |
| $\bar{\mu}$ | Characteristic shear modulus | $1 \times 10^{11} \ \mathrm{Pa}$ |
| $\eta$ | Non-dimensional viscosity | - |
| $\bar{\eta}$ | Characteristic viscosity | $1 \times 10^{21} \ \mathrm{Pa \, s}$ |
| $\alpha$ | Non-dimensional Maxwell time | $\frac{\eta}{\mu}$ |
| $\bar{\alpha}$ | Characteristic time scale | $\frac{\bar{\eta}}{\bar{\mu}} = 1 \times 10^{10} \ \mathrm{s}$ |
| $g$ | Non-dimensional gravitational acceleration | - |
| $\bar{g}$ | Characteristic gravitational acceleration | $9.81 \ \mathrm{m/s^2}$ |
| $\hat{\boldsymbol{e}}_k$ | Radial unit vector | - |
| $B_\mu$ | Non-dimensional ratio of buoyancy to elastic shear strength | $\frac{\bar{\rho}\bar{g}L}{\bar{\mu}} \approx 1.3$ |
| $h_{\mathrm{load}}$ | Non-dimensional load thickness | - |
| $\Delta t$ | Timestep size | - |
| $\boldsymbol{\phi}$ | Finite element test function | - |
| $\hat{\boldsymbol{n}}$ | Outward pointing surface normal vector | - |





For a linear, compressible viscoelastic material, the non-dimensional constitutive equation takes the form

$$\boldsymbol{\sigma}_1^L = \kappa \nabla \cdot \boldsymbol{u}(t)\,\boldsymbol{I} + 2\mu_0 \boldsymbol{d}(t) - 2\sum_i \mu_i \boldsymbol{m}_i(t), \tag{10}$$

where $\kappa$ is the non-dimensional bulk modulus and $\mu_0$ is the non-dimensional effective shear modulus given by

$$\mu_0 = \sum_i \mu_i \tag{11}$$

where $\mu_i$ are the non-dimensional shear moduli associated with each internal variable, $\boldsymbol{m}_i$. The deviatoric strain tensor is given as

$$\boldsymbol{d} = \boldsymbol{e} - \frac{1}{3}\mathrm{Tr}(\boldsymbol{e})\boldsymbol{I}, \tag{12}$$

where $\mathrm{Tr}(\cdot)$ is the trace operator and the strain tensor, $\boldsymbol{e}$, is

$$\boldsymbol{e} = \frac{1}{2}\left(\nabla \boldsymbol{u} + (\nabla \boldsymbol{u})^T\right). \tag{13}$$

We note that all non-dimensional moduli are obtained by division of terms by the characteristic shear modulus $\bar{\mu}$ (Table 1). Each internal variable, $\boldsymbol{m}_i$, is defined by

$$\boldsymbol{m}_i = \frac{1}{\alpha_i}\int_{t_0}^{t} \mathrm{e}^{-\frac{(t-t')}{\alpha_i}}\,\boldsymbol{d}(t')\,dt', \tag{14}$$

where $\alpha_i$ is the non-dimensional Maxwell time for each element. Equivalently, each internal variable evolves according to

$$\partial_t \boldsymbol{m}_i + \frac{1}{\alpha_i}\left(\boldsymbol{m}_i - \boldsymbol{d}\right) = \boldsymbol{0}, \quad \boldsymbol{m}_i(t_0) = \boldsymbol{0}. \tag{15}$$

This formulation provides a compact, flexible and convenient means to incorporate transient rheology into viscoelastic deformation models: using a single internal variable is equivalent to a simple Maxwell material; two correspond to a Burgers model with two characteristic relaxation frequencies; and using a series of internal variables permits approximation of a continuous range of relaxation timescales for more complicated rheologies, including transience. This formulation is also readily extensible to non-linear constitutive equations (e.g., Crawford et al., 2017).

## 2.2   Boundary conditions

To complete the problem specification, we must also define the boundary conditions. At the top of the domain (i.e., Earth's surface), normal stress is balanced by the applied surface load such that

$$\hat{\boldsymbol{n}} \cdot \boldsymbol{\sigma}_1^L = -B_\mu\,\rho_{\mathrm{load}}\,g\,h_{\mathrm{load}}\,\hat{\boldsymbol{e}}_k, \tag{16}$$

where $\rho_{\mathrm{load}}$ and $h_{\mathrm{load}}$ are the non-dimensional density and vertical thickness of the surface load, and $\hat{\boldsymbol{n}}$ is the outward unit
normal vector to the boundary. At the base of the domain (i.e., core–mantle boundary), we impose a no-normal-displacement condition

$$\boldsymbol{u} \cdot \hat{\boldsymbol{n}} = 0, \tag{17}$$



which is consistent with the approximation of a rigid core following Weerdesteijn et al. (2023). Note that this choice departs from models that treat the core as an inviscid fluid and instead impose a free-surface condition (e.g., Zhong et al., 2003; Latychev et al., 2005). After the introduction of self-gravity this should be a trivial extension to G-ADOPT.

Finally, we assume continuity of both displacement and traction across all internal boundaries, such that

$$[\boldsymbol{u}]_-^+ = \boldsymbol{0}, \tag{18}$$

$$[\hat{\boldsymbol{n}} \cdot \boldsymbol{\sigma}_{L1}]_-^+ = \boldsymbol{0}, \tag{19}$$

where $[\cdot]_-^+$ denotes the jump in the associated property across an interface.

In summary, the governing equations for viscoelastic loading consist of the conservation of momentum (Eq. 9) driven by the evolving surface load (Eq. 16) along with the conservation of mass (Eq. 4) and a viscoelastic constitutive equation (Eqs. 10 and 15). Note that this linearised formulation is strictly relevant for small displacements with respect to the depth of the mantle and that the background stress field is assumed to be in hydrostatic equilibrium, implying no lateral variations in density (Al-Attar and Tromp, 2014). For simplicity, we have also neglected rotational and self-gravitational effects in this study. Nevertheless, they are particularly important when coupling the evolving ice and ocean loads through the sea-level equation and will be incorporated in future work (Kendall et al., 2005).

### 2.3 Weak form and spatial discretisation

To derive the finite-element discretisation of these governing equations, we first translate them into their weak form. By selecting appropriate function spaces that contain both solution fields and test functions, the weak form can be obtained by multiplying the equations by their test functions and integrating over the domain, $\Omega$. For conservation of momentum, we use the (vector) test function, $\boldsymbol{\phi}$, to give

$$0 = \int_{\Omega} \boldsymbol{\phi} \cdot \left( \nabla \cdot \boldsymbol{\sigma}_1^L - B_\mu \nabla (\rho_0 g u_k) - B_\mu \rho_1 g \, \hat{\boldsymbol{e}}_k \right) dx, \tag{20}$$

which we then simplify by multiplying both sides by -1 and splitting into three components

$$0 = S + H + D, \tag{21}$$

where

$$S = \int_{\Omega} -\boldsymbol{\phi} \cdot \nabla \cdot \boldsymbol{\sigma}_1^L \, dx, \tag{22}$$

$$H = \int_{\Omega} \boldsymbol{\phi} \cdot B_\mu \nabla (\rho_0 g u_k) \, dx, \tag{23}$$

and

$$D = \int_{\Omega} \boldsymbol{\phi} \cdot B_\mu \rho_1 g \, \hat{\boldsymbol{e}}_k \, dx. \tag{24}$$





A this stage, it becomes necessary to define the finite element spaces that will be used for spatial discretisation. Generally, for each component of displacement, we use $Q2$ finite elements on hexahedral meshes (i.e., the piecewise continuous tri-quadratic tensor product of quadratic continuous polynomials in each direction). In Section 3.2, however, we instead use prismatic meshes that are unstructured in the horizontal but layered in the vertical (facilitated by Firedrake's inbuilt 'extruded' mesh

functionality, this space uses the piecewise continuous bi-quadratic tensor product of a quadratic polynomial in the horizontal and a quadratic polynomial in the vertical; Bercea et al., 2016; McRae et al., 2016). For the deviatoric strain tensor (and hence the internal variable), since it is proportional to the gradient of displacement, we choose the discontinuous $DG1$ space (i.e., linear variations within each finite element cell and discontinuous jumps between cells) for each component. For purely radial variations in density, viscosity and shear modulus, we choose the $DG0$ space (i.e., constant within a finite element cell but

discontinuous between cells), while for laterally varying viscosity fields, we again select $DG1$ finite element functions.

Once these spaces have been chosen, we can integrate the divergence of the incremental Lagrangian stress term by parts within each element, $K_n$, to introduce weak boundary conditions and to move derivatives from the trial function to the test function according to

$$S = \sum_n \left( \int_{K_n} \nabla \boldsymbol{\phi} : \boldsymbol{\sigma}_1^L \, dx - \int_{\partial K_n} \boldsymbol{\phi} \cdot \left( \hat{\boldsymbol{n}} \cdot \boldsymbol{\sigma}_1^L \right) ds \right), \tag{25}$$

where $\hat{\boldsymbol{n}}$ is the outward pointing normal vector to an element edge, represented by $\partial K_n$. Given that test functions are continuous across cell edges and we assume that there is no jump in stress across material discontinuities, Eq. (25) simplifies to

$$S = \int_\Omega \nabla \boldsymbol{\phi} : \left( \kappa \nabla \cdot \boldsymbol{u} \, \boldsymbol{I} + 2\mu_0 \boldsymbol{d}(\boldsymbol{u}) - 2 \sum_i \mu_i \boldsymbol{m}_i(\boldsymbol{u}) \right) dx + \int_{\partial \Omega_{\text{top}}} \boldsymbol{\phi} \cdot (B_\mu \, \rho_{\text{load}} \, g \, h_{\text{load}} \hat{\boldsymbol{n}}) \, ds, \tag{26}$$

where we have substituted Eq. (10) for the constitutive relation and Eq. (16) for the boundary condition at the top of the domain, $\partial \Omega_{\text{top}}$. For bottom (and side) boundaries where we specify no normal displacement (i.e., Eq. 17), we utilise two different

strategies depending on domain geometry. In Cartesian domains, we apply the boundary condition strongly by modifying the discrete test and trial spaces, such that the surface integral vanishes. For spherical domains, we implement that condition weakly using the Symmetric Interior Penalty Galerkin method (Epshteyn and Rivière, 2007; Hillewaert, 2013). An example showing implementation of this approach for mantle convection problems in Firedrake can be seen in Davies et al. (2022).

The same approach is taken for advection of the hydrostatic pre-stress term, initially integrating by parts to give

$$H = \sum_i \left( \int_{K_i} -\nabla \cdot \boldsymbol{\phi} \, B_\mu \rho_0 g u_k \, dx + \int_{\partial K_i} \boldsymbol{\phi} \cdot \hat{\boldsymbol{n}} B_\mu \rho_0 g u_k \, ds \right). \tag{27}$$

Since we allow density and gravity discontinuities through the $DG0$ discretisation, we need to also account for the contribution of surface integrals along interior facets, denoted by $\Gamma$, using

$$H = -\int_\Omega \nabla \cdot \boldsymbol{\phi} B_\mu \rho_0 g u_k \, dx + \int_\Gamma \boldsymbol{\phi} \cdot B_\mu u_k [[\rho_0 g \hat{\boldsymbol{n}}]] \, ds + \int_{\partial \Omega_{\text{top}}} \boldsymbol{\phi} \cdot \hat{\boldsymbol{n}} B_\mu \rho_0 g u_k \, ds, \tag{28}$$



where the jump term is given by

$$[[\rho_0 g\hat{\boldsymbol{n}}]] = \hat{\boldsymbol{n}}^+ \rho_0^+ g^+ + \hat{\boldsymbol{n}}^- \rho_0^- g^- . \tag{29}$$

The (arbitrary) labels $+$ and $-$ mark contributions from either side of the cell edge and $\hat{\boldsymbol{n}}^+ = -\hat{\boldsymbol{n}}^-$. Note that the interior facet term is only non zero across layers with material density and gravity jumps and is similar to the Winkler foundations as described in Wu (2004). We also note that in the incompressible formulations of Zhong et al. (2003) and Wu (2004), the first term of Eq. (28) can instead be integrated within a pressure gradient term. Finally, the last term in Eq. (28) is similar to a free surface feedback term encountered in mantle convection (Kramer et al., 2012).

## 2.4 Time discretisation

We now discretise in time using the implicit Backward Euler (BE) scheme. This choice allows us to take timesteps larger than the characteristic Maxwell time without compromising numerical stability. Such flexibility is particularly advantageous when the timescale of glacial loading is substantially slower than the Maxwell time – as is often the case in low-viscosity regions – thereby avoiding having to take prohibitively small timesteps in realistic simulations of glacial cycles (e.g. Bailey, 2006; Lloyd et al., 2024). Applying the BE scheme, the evolution of each internal variable in Eq. (15) becomes

$$\boldsymbol{m}_i^{n+1} = \frac{1}{1 + \frac{\Delta t}{\alpha_i}} \left( \boldsymbol{m}_i^n + \frac{\Delta t}{\alpha_i} \boldsymbol{d}(\boldsymbol{u}^{n+1}) \right), \tag{30}$$

where the superscript $n$ refers to the previous timestep, $n+1$ is the next timestep, and $\Delta t$ is the timestep duration. We then substitute this result into Eq. (26), yielding

$$S = \int_\Omega \nabla\boldsymbol{\phi} : \left( \kappa \nabla\cdot\boldsymbol{u}^{n+1}\boldsymbol{I} + 2\sum_i \frac{\eta_i}{\alpha_i + \Delta t} \left( \boldsymbol{d}(\boldsymbol{u}^{n+1}) - \boldsymbol{m}_i^n \right) \right) dx + \int_{\partial\Omega_{\text{top}}} \boldsymbol{\phi}\cdot(B_\mu\,\rho_{\text{load}}\,g\,h_{\text{load}}\hat{\boldsymbol{n}})\,ds \tag{31}$$

where $\eta_i$ is the non-dimensional viscosity of the the internal variable $\boldsymbol{m}_i$. Recombining Eqs. (24), (28) and (31), the final system of equations is

$$\int_\Omega \nabla\boldsymbol{\phi} : \left( \kappa\nabla\cdot\boldsymbol{u}^{n+1}\boldsymbol{I} + 2\sum_i \frac{\eta_i}{\alpha_i + \Delta t}\boldsymbol{d}(\boldsymbol{u}^{n+1}) \right) dx$$

$$-\int_\Omega \nabla\cdot\boldsymbol{\phi}\,B_\mu\rho_0 g u_k^{n+1}\,dx + \int_\Gamma \boldsymbol{\phi}\cdot B_\mu u_k^{n+1}[[\rho_0 g\hat{\boldsymbol{n}}]]\,ds + \int_{\partial\Omega_{\text{top}}} \boldsymbol{\phi}\cdot\hat{\boldsymbol{n}}B_\mu\rho_0 g u_k^{n+1}\,ds$$

$$-\int_\Omega \boldsymbol{\phi}\cdot B_\mu g\left(\boldsymbol{u}^{n+1}\cdot\nabla\rho_0 + \rho_0\nabla\cdot\boldsymbol{u}^{n+1}\right)\hat{\boldsymbol{e}}_k\,dx$$

$$= -\int_{\partial\Omega_{\text{top}}} \boldsymbol{\phi}\cdot(B_\mu\,\rho_{\text{load}}\,g\,h_{\text{load}}\hat{\boldsymbol{n}})\,ds + \int_\Omega \nabla\boldsymbol{\phi} : \left( 2\sum_i \frac{\eta_i}{\alpha_i + \Delta t}\boldsymbol{m}_i^n \right) dx, \tag{32}$$

where implicit terms involving the unknown displacement at the next time step, $\boldsymbol{u}^{n+1}$, have been collected on the left-hand side and explicit terms known from the current time step are on the right-hand side. Finding the new values of $\boldsymbol{u}^{n+1}$ requires





solving a linear system at each timestep, which is solved in Firedrake through PETSc's comprehensive linear algebra library

(Balay et al., 1997, 2025a, b). To deal with non-symmetric terms in lines 2 and 3 of Eq. (32), we employ *GMRES* as the Krylov method. We also choose PETSc's algebraic multigrid package, *GAMG*, as a preconditioner since it has proven to be a robust and effective method for other large-scale problems in geodynamics (e.g. Davies et al., 2022).

In testing, we have found that the ratio of bulk-to-deviatoric strain terms in line 1 of Eq. (32) plays a crucial role in determining solver performance. In particular, when the bulk strain term dominates, the resulting system can become poorly conditioned

and leads to an increased number of iterations. This issue is exacerbated when timesteps greatly exceed the Maxwell time, as the relative contribution of the deviatoric strain term is reduced in that scenario. Since the ratio of bulk-to-shear modulus is typically modest (i.e., $\sim 2$) for mantle rocks, we do not expect this behaviour to be problematic when using our software to run realistic GIA simulations in future work. Nevertheless, when undertaking incompressible benchmark tests, it does increase the computational cost of attempting to approximate incompressible materials by raising this ratio. An alternative, more common

approach in the incompressible limit is to reintroduce pressure as a Lagrange multiplier to enforce the conservation of volume constraint $\nabla \cdot \boldsymbol{u} = 0$ (e.g. Zhong et al., 2003). The resulting saddle-point system can be effectively solved using a Schur complement factorisation strategy (e.g. Davies et al., 2022), which we have implemented during code development but do not discuss further here since it is less applicable to real-Earth simulations.

One way to mitigate timestep sensitivity is to reformulate the system, such that the internal variable evolution equation is

solved simultaneously with the momentum equation, yielding a coupled system for $\boldsymbol{u}^{n+1}$ and $\boldsymbol{m}^{n+1}$. While this approach incurs some additional outer iterations to maintain coupling, it also ensures that convergence of the momentum solve is no longer dependent on timestep size. Under this formulation, the momentum equation becomes

$$\int_\Omega \nabla\boldsymbol{\phi} : \left( \kappa \nabla \cdot \boldsymbol{u}^{n+1}\,\boldsymbol{I} + 2\mu_0 \boldsymbol{d}(\boldsymbol{u}^{n+1}) \right) dx$$

$$- \int_\Omega \nabla \cdot \boldsymbol{\phi} B_\mu \rho_0 g u_k^{n+1}\, dx + \int_\Gamma \boldsymbol{\phi} \cdot B_\mu u_k^{n+1} [[\rho_0 g \hat{\boldsymbol{n}}]]\, ds + \int_{\partial\Omega_{\text{top}}} \boldsymbol{\phi} \cdot \hat{\boldsymbol{n}} B_\mu \rho_0 g u_k^{n+1}\, ds$$

$$- \int_\Omega \boldsymbol{\phi} \cdot B_\mu g \left( \boldsymbol{u}^{n+1} \cdot \nabla\rho_0 + \rho_0 \nabla \cdot \boldsymbol{u}^{n+1} \right) \hat{\boldsymbol{e}}_k\, dx$$

$$= - \int_{\partial\Omega_{\text{top}}} \boldsymbol{\phi} \cdot \left( B_\mu\, \rho_{\text{load}}\, g\, h_{\text{load}}\, \hat{\boldsymbol{n}} \right) ds + \int_\Omega \nabla\boldsymbol{\phi} : \left( 2\sum_i \mu_i \boldsymbol{m}^{n+1} \right) dx, \quad (33)$$

and Eq. (30) is converted into its weak form by integrating over the domain and testing with $\boldsymbol{\psi}$, a (tensor) $DG1$ finite element test function (i.e., linear variation within cells, discontinuous between them), yielding

$$\int_\Omega \boldsymbol{\psi} \cdot \boldsymbol{m}_i^{n+1}\, dx = \int_\Omega \boldsymbol{\psi} \cdot \left( \frac{1}{1 + \dfrac{\Delta t}{\alpha_i}} \left( \boldsymbol{m}_i^n + \frac{\Delta t}{\alpha_i} \boldsymbol{d}(\boldsymbol{u}^{n+1}) \right) \right) dx. \tag{34}$$

This coupled formulation can be solved for $\boldsymbol{u}^{n+1}$ and $\boldsymbol{m}_i^{n+1}$ using the *GMRES* Krylov method at the outer level, combined with a block Gauss-Seidel approach. Within the inner solve, Eq. (33) is handled via the same *GMRES* and *GAMG* strategy as



before, while the computationally simpler Eq. (34) can be solved using the conjugate gradient method, preconditioned with successive over-relaxation. While the coupled system offers greater robustness in cases involving either long timesteps or large bulk-to-shear modulus ratios, for more typical mantle bulk-to-shear ratios, the substitute method (i.e., Eq. 32) is generally more
computationally efficient as it avoids additional outer iterations. We note that in all results that follow, unless explicitly stated otherwise, the substitution approach is utilised.

To summarise this section, we have outlined the viscoelastic loading equations and discretisation as currently implemented in G-ADOPT. In particular, by adopting the internal variable formulation of viscoelasticity (Al-Attar and Tromp, 2014; Crawford et al., 2017, 2018), we are able to accommodate a broad class of complex rheologies with relative ease in comparison to many
of the traditional, Maxwell-based approaches. Additionally, our implementation of an implicit timestepping scheme enables stable integration even when timesteps are significantly longer duration than the Maxwell time. This flexibility should prove particularly advantageous in longer glacial loading scenarios, where low-viscosity regions that have Maxwell times on the order of a year would otherwise impose prohibitively small timestep constraints.

## 3  Numerical Experiments and Results

We next undertake a series of benchmarking experiments to assess the accuracy, robustness, and flexibility of our code.

### 3.1  Analytical comparisons

In this section, we assess the accuracy of G-ADOPT in solving simple viscoelastic loading problems that have analytical solutions for both compressible and incompressible cases. Such benchmarks provide a stringent check as they also allow us to test whether, as the temporal and spatial resolution is refined, the error decreases at the expected theoretical rate.

For the first scenario, we consider the vertical surface displacement, $u_r$, of a 2D Maxwell viscoelastic half-space that is subjected to emplacement of an instantaneous, sinusoidal surface load. By rearranging Eq. 2b from Cathles (2024), we can simulate a loading, rather than an unloading scenario. The analytical solution is

$$u_r = F_0 \left( 1 - \frac{1}{1 + f_e \alpha / \tau} + \frac{1 - e^{-t/(\tau + f_e \alpha)}}{1 + f_e \alpha / \tau} \right) \cos(kx), \tag{35}$$

where $F_0$ is the load thickness (where its density is equal to the mantle density), $\alpha = \frac{\eta}{\mu}$ is the Maxwell time, $\eta$ is the viscosity,
$\mu$ is the shear modulus, and $\tau$ is the viscous relaxation timescale given by

$$\tau = \frac{2k\eta}{\rho g}, \tag{36}$$

where $k$ is the wavenumber, $\rho$ is the density, and $g$ is the gravitational acceleration. The compressibility parameter, $f_e$, is defined as

$$f_e = \frac{\lambda_e + 2\mu}{\lambda_e + \mu}, \tag{37}$$

where $\lambda_e$ is the (first) elastic Lamé parameter, given by

$$\lambda_e = \kappa - \frac{2\mu}{3}, \tag{38}$$



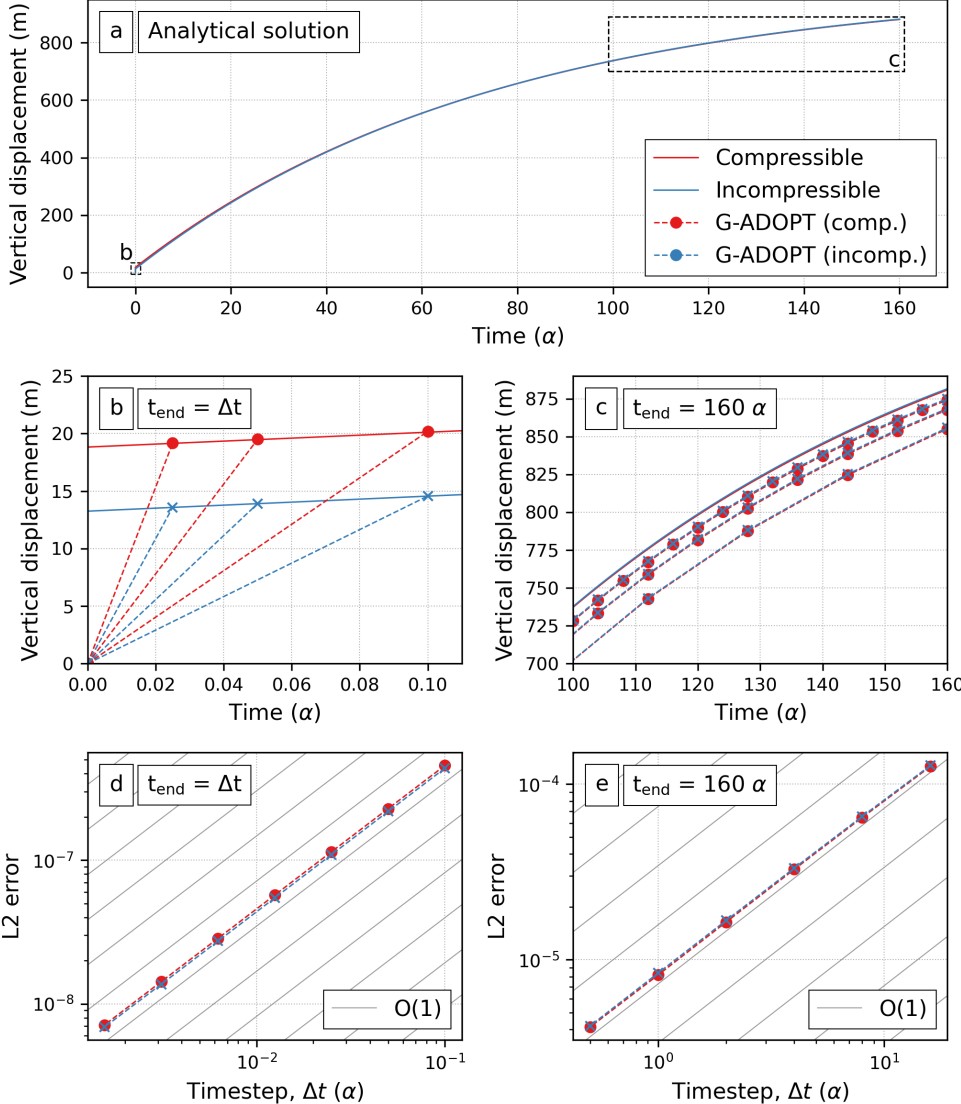

**Figure 1.** Analytical comparisons for 2D viscoelastic loading problem. (a) Peak displacement vs. time for compressible (red) and incompressible (blue) cases. Highlighted boxes indicate the time-windows shown in panels (b) and (c) and the legend applies to the whole figure. (b) Elastic-limit comparison of G-ADOPT predictions (dashed) with analytical solutions (solid) at three timesteps ($\Delta t = 0.1\alpha$, $0.05\alpha$, and $0.025\alpha$). (c) Viscous-limit comparison to $t = 160\alpha$ for $\Delta t = 16\alpha$, $8\alpha$ and $4\alpha$. (d) $L_2$-norm of the surface displacement error in the elastic limit as a function of timestep size ($\Delta t = 0.1\alpha$ to $0.0015625\alpha$), demonstrating first-order convergence. (e) Time-integrated global $L_2$-norm error in the viscous limit as timestep size is refined ($\Delta t = 16\alpha$ to $0.5\alpha$). Diagonal lines in (d) and (e) indicate expected first-order convergence behaviour.





where $\kappa$ is the bulk modulus. When the material is incompressible, then $f_e \to 1$.

For consistency with the analytical solution, we must neglect both compressible buoyancy and advection of hydrostatic pre-stress, which appear in Eqs. (4) and (28). Feedback from the free surface, given by the last term in Eq. (28), is retained to ensure correct asymptotic behaviour in the viscous limit.


We adopt a mantle viscosity of $\eta = 1 \times 10^{21}$ Pa s and shear modulus of $\mu = 1 \times 10^{11}$ Pa, yielding a Maxwell time of $\alpha \approx 320$ years. For the compressible case, the bulk modulus is $\kappa = 2 \times 10^{11}$ Pa, yielding $f_e = 1.43$. Density and gravitational acceleration are set to 4500 kg m$^{-3}$ and 10 m s$^{-2}$, respectively. The applied load has a height of 1 km and wavelength of 375 km (equivalent to $\frac{1}{8}$ of the domain depth), leading to a viscous relaxation time of $\tau \approx 24,000$ years.


Figure 1a shows the temporal evolution of peak surface displacement in the analytical solutions over a duration of 160 Maxwell times (i.e., approximately 2.2 viscous relaxation timescales). Both compressible and incompressible cases exhibit similar behaviour: an immediate elastic displacement is followed by a gradual approach towards isostatic equilibrium. Including compressibility leads to a slightly larger initial elastic deformation ($\sim 19$ m, as opposed to $\sim 13$ m; Figure 1b), but the viscous asymptote is the same for both scenarios (Figure 1c).


To explore numerical accuracy as a function of timestep size, we fix the mesh resolution and perform a suite of simulations over a range of timestep durations. We use a 2D domain of depth 1 and width 0.5 in non-dimensional terms that generally contains $320 \times 320$ cells, except in the incompressible, long-duration case where this resolution was doubled to $640 \times 640$ cells to ensure that the time discretisation error dominates over any spatial discretisation errors. Since the Backward Euler timestep method is unconditionally stable and first-order accurate, these tests should ideally achieve a first-order convergence rate.


At short timescales where elastic deformation is expected to dominate, we undertake a single timestep with durations of $\Delta t = 0.1\alpha$ down to $0.0015625\alpha$. To approximate incompressibility within our compressible viscoelastic formulation, we set the bulk modulus to a high value of $\kappa = 1 \times 10^{15}$ Pa (equivalent to $f_e = 1.0001$). For both compressible and incompressible cases, as $\Delta t$ reduces, we can see that our viscoelastic response gets systematically closer to the instantaneous elastic response from the analytical solutions at time $t = 0$ (Figure 1b) and that the surface $L_2$-errors of this difference reduce with first-order accuracy (Figure 1d).


At long timescales where viscous deformation dominates, we run numerical simulations with a timestep duration that systematically varies from 16 $\alpha$ down to 0.5 $\alpha$. Since the relative contribution of elastic compressibility to total displacement is much lower over long timescales, we use a less extreme bulk modulus of $\kappa = 1 \times 10^{13}$ Pa for the incompressible case (i.e., $f_e = 1.01$). Once again, we observe that, as timestep size reduces, numerical simulations approach the two analytical solutions (Figure 1c) and that their time-integrated global $L_2$-error reduces with first-order convergence (Figure 1e).


Despite the physical simplicity necessitated by these analytical solutions, the results demonstrate that G-ADOPT accurately reproduces the analytical response for both compressible and incompressible Maxwell halfspaces and that its implicit time-stepping scheme achieves the expected first-order accuracy in time. In the following benchmarks, we next evaluate G-ADOPT against other finite element and semi-analytical codes using more realistic loading scenarios in a 3D domain.





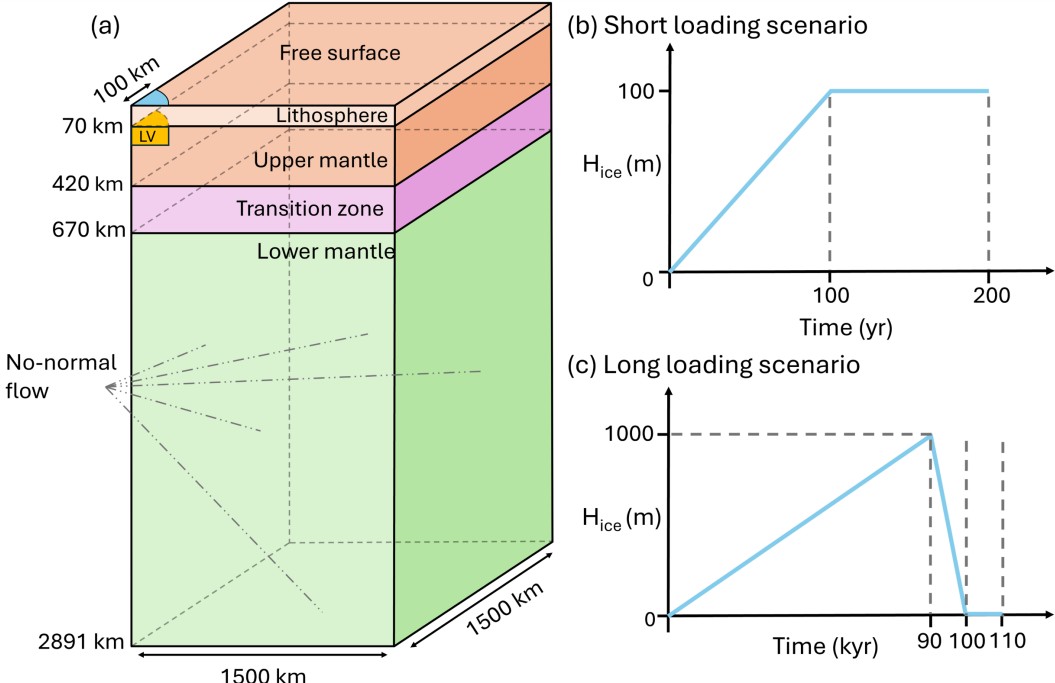

**Figure 2.** Schematic of the 3D Cartesian benchmark from Weerdesteijn et al. (2023). (a) Model domain showing rheological layers, an optional low viscosity (LV) zone, and the extent of ice at the surface. Note schematic is not drawn to scale. (b) Temporal evolution of thickness of the ice disk in the short-loading scenario. (c) Same for the long-loading scenario.

## 3.2 3D Cartesian benchmarks with a Maxwell rheology

Weerdesteijn et al. (2023) have recently presented a valuable suite of test simulations created using three published GIA models: two finite element codes – Abaqus (Wu, 2004) and Aspect (Weerdesteijn et al., 2023) – and one semi-analytical code that assumes an axisymmetric spherical Earth, Taboo (Spada et al., 2004). A schematic of their model setup is shown in Figure 2, for which the domain is a 3D Cartesian box measuring $1500 \times 1500$ km laterally and 2891 km in depth (and therefore can only be approximately represented in Taboo). Two loading scenarios were considered: (i) a short-duration event that has similar spatiotemporal scales to modern ice changes, whereby a 100 m-thick ice sheet linearly grows for 100 years before being held constant for another 100 years; and (ii) a long-duration scenario that is more similar to a glacial cycle, whereby 1 km of ice linearly accumulates over 90 kyr and then fully melts by 100 kyr, followed by a further 10 kyr of evolution in the absence of further loading changes. In both cases, the ice sheet is represented by a disc with a radius of 100 km.

The default Earth structure follows the 1D layered model of Spada et al. (2011), composed of a lithosphere, upper mantle, transition zone, and lower mantle (Table 2). To assess sensitivity to lateral viscosity variations, a second 3D-Earth structure also incorporates a cylindrical anomaly with a radius of 100 km and a reduced viscosity of $1 \times 10^{19}$ Pa s directly beneath the




**Table 2.** Rheological parameters for 1D depth profiles following Weerdesteijn et al. (2023) and Spada et al. (2011). To approximate infinite viscosities in the lithosphere, our simulation assumes $\eta = 1 \times 10^{40}$ Pa s.

| Layer | Radius (km) | Density (kg m$^{-3}$) | Shear modulus (Pa) | Viscosity (Pa s) | Gravity (m s$^{-2}$) |
|---|---|---|---|---|---|
| Lithosphere | 6371 | 3037 | $0.50605 \times 10^{11}$ | $\infty$ | 9.815 |
| Upper mantle | 6301 | 3438 | $0.70363 \times 10^{11}$ | $1 \times 10^{21}$ | 9.854 |
| Transition zone | 5951 | 3871 | $1.05490 \times 10^{11}$ | $1 \times 10^{21}$ | 9.978 |
| Lower mantle | 5701 | 4978 | $2.28340 \times 10^{11}$ | $2 \times 10^{21}$ | 10.024 |
| Core | 3480 | 10750 | 0 | 0 | 10.457 |

ice sheet, extending from 70–170 km depth. Since Taboo assumes spherical symmetry and is therefore limited to modelling with 1D profiles, only Abaqus and Aspect provide 3D results that can be directly compared with G-ADOPT.

The configuration in G-ADOPT is set up to closely imitate Aspect's implementation of the benchmark, which includes: neglecting self-gravity; using a constant gravitational acceleration of 9.815 m s$^{-2}$; setting viscosity in the elastic lithosphere to $1 \times 10^{40}$ Pa s (thereby ensuring negligible viscous deformation of this layer over the course of the simulation); and exploiting axisymmetry to model only one quarter of the domain with no-normal-flow boundary conditions imposed on its side walls and base. Ice loading is implemented using a smooth hyperbolic tangent profile with a 1 km 'rollover' length-scale and the density

of ice is assumed to be 931 kg m$^{-3}$. We use a horizontally unstructured mesh, refined to 5 km under the ice and coarsened to 200 km in the far field. Vertically, each rheological layer is discretised with a fixed number (e.g., 10 per layer in the default setup) of $DG0$ elements, providing constant values within elements but permitting sharp boundaries between layers. Timesteps are 10 and 1000 years for the short- and long-loading scenarios, respectively. For reference, Abaqus uses a similar meshing approach to G-ADOPT, with horizontal resolution of 5–200 km and 8 vertical cells per rheological layer; Aspect uses $\sim 6$ km

isotropic resolution in the top 100 km of the domain, transitioning to 50 km elsewhere; and Taboo uses a maximum spectral degree of 4096, equivalent to approximately 5 km horizontal resolution, and is semi-analytical in the vertical (Weerdesteijn et al., 2023). Since the benchmarks all assume incompressibility, we again approximate incompressibility by setting the bulk modulus to $1 \times 10^{14}$ Pa (i.e., 1000 times larger than the reference shear modulus). Note that, to ensure robustness in these incompressible simulations, we solve the coupled system (i.e., Eqs. 33 and 34) for reasons outlined in Section 2.

Figure 3 compares maximum vertical displacements beneath the ice load across all four codes in each of the four benchmark combinations of Earth model and loading scenario. The upper row shows results for 1D Earth structure, while the lower row includes the low-viscosity cylindrical anomaly. Agreement is generally excellent in all cases where cross-code comparisons are possible (Figures 3a–3c), affirming consistency with G-ADOPT's implementation. In the final panel (Figure 3d), we present results from G-ADOPT for the long-loading scenario with 3D viscosity – a configuration not illustrated in Weerdesteijn et al.

(2023). In comparison to the same loading scenario on a 1D Earth, the presence of a low-viscosity patch accelerates the surface response consistent with expectations, but only modestly: the maximum vertical displacement differs by just $\sim 2\%$ relative to the 1D case at 90 kyr, compared to a $\sim 60\%$ difference in the short-duration loading scenario. This result is consistent with the





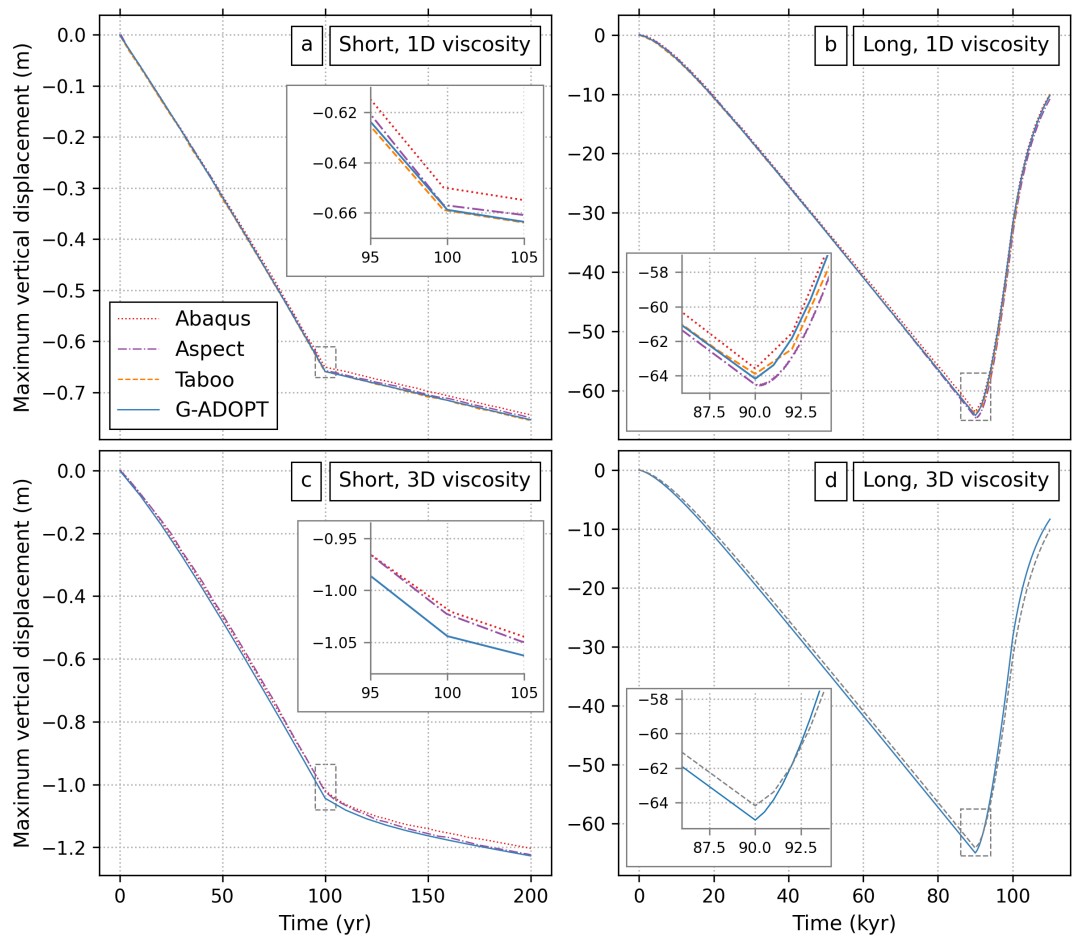

**Figure 3.** Comparison of maximum vertical surface displacements beneath the ice load for the 3D Cartesian box benchmark scenarios of Weerdesteijn et al. (2023). (a) Short-duration loading scenario on a 1D Earth model. Dotted red = Abaqus; dash-dotted purple = Aspect; dashed orange = Taboo; and solid blue = G-ADOPT. Inset shows a zoom of the region outlined by the black-dashed box. (b) Same for the long-duration loading scenario on a 1D Earth. (c) Short-duration loading scenario on a 3D Earth containing a low-viscosity patch in the upper mantle beneath the ice sheet. (d) Same for the long-duration loading scenario on a 3D Earth; grey dashed line = G-ADOPT result on a 1D Earth (panel b) plotted to highlight impact of the low-viscosity patch. Note that results for Abaqus, Aspect and Taboo are digitised from Weerdesteijn et al. (2023) and that not all scenarios were tested in those codes. Also note the different different displacement extent on each y-axis.



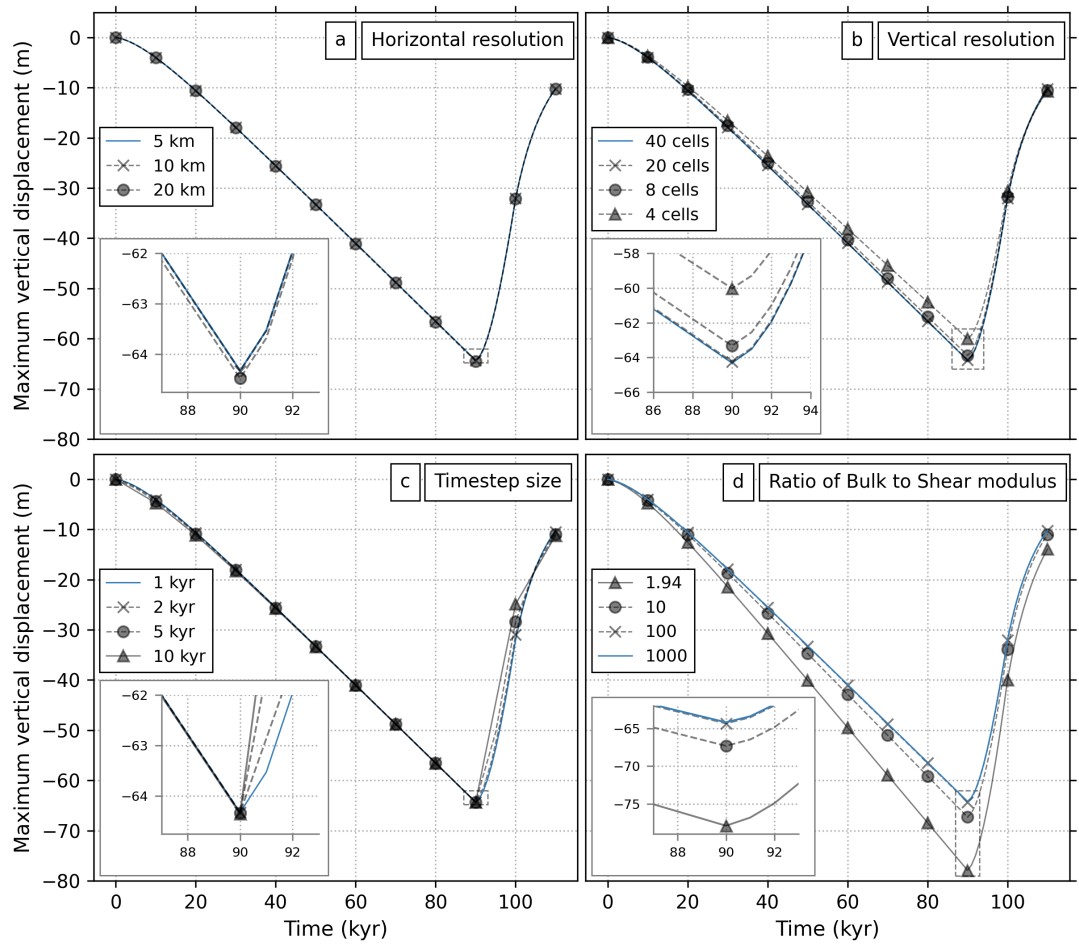

**Figure 4.** Sensitivity analysis for G-ADOPT using the scenario explored in Figure 3b consisting of long-duration ice loading on a 1D Earth model. (a) Impact of horizontal resolution. Solid blue = default spacing (5 km beneath ice sheet, 200 km in far field); crosses = half of default resolution; circles = one quarter of default resolution. (b) Impact of vertical resolution. Solid blue = default case with 40 vertical cells across the model domain (i.e., 10 per layer); crosses = 20 cells; circles = eight cells; triangles = four cells. (c) Impact of timestep size. Solid blue = default of 1 kyr; crosses = 2 kyr; circles = 5 kyr; triangles = 10 kyr. (d) Impact of transitioning from incompressible to a realistic Earth-like level of compressibility by modifying the ratio of bulk-to-shear modulus. Solid blue = 1000; crosses = 100; circles = 10; triangles = 1.94.

fact that, even for the 1D Earth model, the $\sim 100$ kyr loading timescale is substantially longer than the $\sim 320$ year Maxwell time of the upper mantle (N.B., the equivalent Maxwell time for the low-viscosity patch is only $\sim 3$ years).

To further assess the robustness of G-ADOPT, we conduct a resolution and parameter sensitivity analysis for the scenario involving long-duration loading of a 1D Earth model. Horizontal mesh sensitivity tests demonstrate that even as few as five elements beneath the ice-covered region (i.e., 20 km horizontal resolution) are sufficient to accurately capture the pattern of deformation (Figure 4a). Changing the vertical resolution has a bigger effect (Figure 4b). Halving the default values to five




**Table 3.** Weak scaling analysis for the long-duration loading case on a 1D Earth using a structured mesh. Default iteration counts and timings use a bulk-to-shear modulus ratio of 100 (equivalent values for a ratio of 1.94 are given in brackets). DOFs = Degrees of freedom.

| DOFs | Cores | Iterations | Time per timestep (s) |
|---|---|---|---|
| 3168963 | 104 | 184 (27) | 36.3 (9.3) |
| 24961923 | 832 | 209 (30) | 44.3 (12.5) |
| 198147843 | 6656 | 247 (49) | 55.0 (19.1) |

elements per layer (i.e., 20 vertical elements in total) results in practically indistinguishable results, but reducing to two cells per
layer produces peak differences of $\sim 1\%$, which increases to $\sim 6\%$ when only using one cell per layer. Regarding the impact
of timestep size, these tests neatly demonstrate the effectiveness of our implicit time-stepping scheme. Even when the timestep
is increased to 10 kyr, which is more than 30 times greater than the Maxwell time of the upper mantle in these models, the
solution remains stable and accurate throughout the loading phase and only produces some modest errors during the unloading
phase (Figure 4c). For this case, the computational cost per timestep is doubled because the number of outer iterations per
timestep increases, but this still leads to an effective speed up of a factor of five in comparison to the 1 kyr simulation. By way
of comparison, Aspect used significantly shorter timesteps (e.g., 50 years) to achieve comparable results in this benchmark
(Weerdesteijn et al., 2023).

We can also use this test scenario to demonstrate the impact of realistic levels of compressibility by changing the ratio of bulk-
to-shear modulus (Figure 4d). A value of 1000 closely approximating incompressibility and turns out to be indistinguishable
from a value of 100. By the time it is lowered to 1.94, there is a $\sim 20\%$ increase in maximum vertical displacement, reflecting
the larger elastic deformation caused by loading compressible materials. This result further underscores the importance of
including compressibility in GIA simulations (Wolf, 1985; Tanaka et al., 2011; A et al., 2013).

To test the scalability of G-ADOPT, we have undertaken a weak scaling analysis on Gadi (Australia's highest performance
CPU-based supercomputer) using the long-duration loading scenario on a 1D Earth and a structured mesh. We use a number
of cores ranging from 104 up to 6,656 (i.e., from 1 to 64 Intel Xeon Sapphire Rapids nodes) and aim to keep the number of
degrees of freedom per core constant at $\sim 30,000$. The results show that, with a bulk-to-shear modulus ratio of 100, increasing
the problem size by a factor of 64 causes the solve time per timestep to increase by $\sim 30\%$ (Table 3), which is consistent with
previous scaling results for mantle convection modelling within Firedrake (Davies et al., 2022). As discussed in Section 2.4,
for the case with a smaller bulk to shear modulus ratio of 1.94, conditioning of the system improves and the number of iteration
counts is significantly reduced. However, the overall scaling performance is slightly worse with the solve time per timestep
effectively doubling as the problem size increases by a factor of 64.

In summary, for a 3D Cartesian box with potential presence of lateral viscosity variations, G-ADOPT accurately reproduces
benchmark results from published codes. It also exhibits desirable performance characteristics, including efficient timestepping
and stability across a range of resolutions. These results validate the core solver and lay the groundwork for investigating more
complex loading scenarios, including those involving compressibility and more complex rheologies.





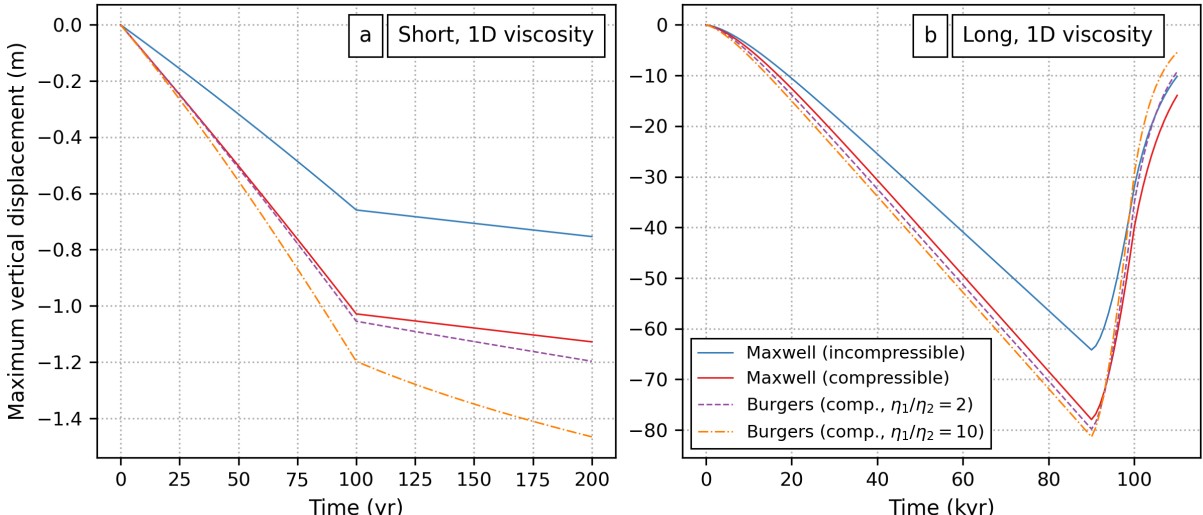

**Figure 5.** Extension of the 3D Cartesian benchmark to a compressible Burgers rheology. (a) Short-duration loading scenario on a 1D Earth model. Solid blue = incompressible Maxwell rheology previously shown in Figure 3a; solid red = equivalent for compressible Maxwell rheology (bulk-to-shear modulus ratio of 1.94) shown in Figure 4d; dashed purple = Burgers rheology with short-term viscosity a factor of 2 lower than the long-term viscosity; dash-dotted orange = same for a factor of 10 viscosity reduction. (b) Same for the long-duration loading scenario.

### 3.3 Extending to complex rheologies including transience

In this section, we demonstrate the flexibility of the internal variable formulation to model compressible, transient rheology. Specifically, we extend the short- and long-duration loading benchmarks on a 1D Earth model to include a compressible, Burgers rheology, which can be conceptually considered as a standard Maxwell body (composed of a spring and dashpot connected in series) connected in series to a single Kelvin-Voigt element (a spring and dashpot connected in parallel). It requires using two internal variables to represent the two characteristic relaxation timescales of the system (Crawford et al., 2017, 2018).

We adopt a bulk-to-shear modulus ratio of 1.94 (corresponding to a Poisson's ratio of 0.28) as that value is standard in Abaqus implementations (Huang et al., 2023). To maintain the same elastic response as in the previous tests of Section 3.2, we halve the shear modulus of each internal variable element, thereby preserving the total effective modulus. Similarly, we halve the viscosity of the long-term viscous element, so that the overall relaxation timescale remains consistent with the Maxwell cases presented earlier. We then explore two different cases in which the viscosity of the element controlling short-term relaxation is set to either one-half or one-tenth that of the long-term element. All other aspects of the setup including grid resolution, timestep size, boundary conditions, and loading configuration are identical to those in Section 3.2.

As discussed before, the impact of compressibility alone is to increase the magnitude of the peak displacement through time in comparison to the incompressible Maxwell case, consistent with expectations. Adding in a transient element with





lower viscosity further amplifies this response, particularly where rapid changes in the load interact strongly with the shorter relaxation timescale (Figure 5). This behaviour is most obvious in the short-duration loading scenario, but it is also visible in the unloading phase of the long-duration scenario. We also verified that setting the same viscosity for both internal variables (i.e., reducing the Burgers model to a single timescale) reproduces the original Maxwell response, confirming the internal consistency of our implementation.

It is noteworthy that, for short-duration loading, displacement curves for the compressible Maxwell case and the compressible Burgers simulations (Figure 5a) exhibit comparable features to those produced for the incompressible Maxwell model with a low-viscosity patch (Figure 3c). Although the underlying deformation mechanisms differ, this similarity illustrates the difficulty of disentangling the effects of local viscosity variations, compressibility and transient rheology (e.g., Pan et al., 2024). Nevertheless, it also highlights the flexibility of the G-ADOPT framework and reinforces its potential future value to these debates.

In summary, we have demonstrated G-ADOPT's ability to model compressible, transient viscoelastic rheologies using the internal variable formulation and the important impact of these phenomena on GIA predictions, particularly during rapid loading changes. In the next section, we demonstrate extension of these simulations to domains with spherical geometry.

## 3.4    Extension to spherical geometries

A significant strength of G-ADOPT is that it is relatively trivial to change the mesh to suite a range of different domain geometries. We demonstrate this aspect by extending our previous simulations from the 3D Cartesian box to a 3D spherical Earth. We adopt a model domain from the benchmark of Spada et al. (2011), which modelled an incompressible Maxwell rheology with purely radial viscosity variations. While we cannot repeat this benchmark as we have not yet implemented self-gravity in G-ADOPT, we have extended it to a more complex rheology by using a compressible, Burgers (i.e., transient) rheology that also contains lateral viscosity variations.

To generate the mesh, we first use Firedrake's inbuilt functionality to create a cubed-sphere surface mesh with 24576 horizontal elements (equivalent to 6 refinements). We then 'extrude' the mesh in the vertical direction to create 40 vertical layers (10 per rheological layer), as in Section 3.2. A couple of additional changes with respect to the Cartesian setup are that we provide a rotational null space to the Firedrake solver options to improve iterative solver performance and provide a weak implementation of the no-normal-flow boundary condition on the core-mantle boundary. We retain the same radial profiles of density, shear modulus, and long-term (i.e., steady-state) viscosity as in Spada et al. (2011), but these viscosities are further modulated by a ($\ell = 4, m = 1$) spherical harmonic function that introduces lateral viscosity variations spanning two orders of magnitude (Figure 6). For the short-term viscosity in the Burgers model, we assign a value that is a factor of 10 times smaller, while compressibility is set using a bulk-to-shear modulus ratio of 1.94, corresponding to a Poisson's ratio of approximately 0.28. A 1 km-thick ice disc with a radius of 10° latitude is placed on the North Pole and applied instantaneously at the start of the simulation via a Heaviside step function. Ice density is 931 kg m$^{-3}$ and gravity is constant: $g = 9.815$ m s$^{-2}$. A timestep of 50 years is used to resolve rapid deformation following the instantaneous ice loading event.



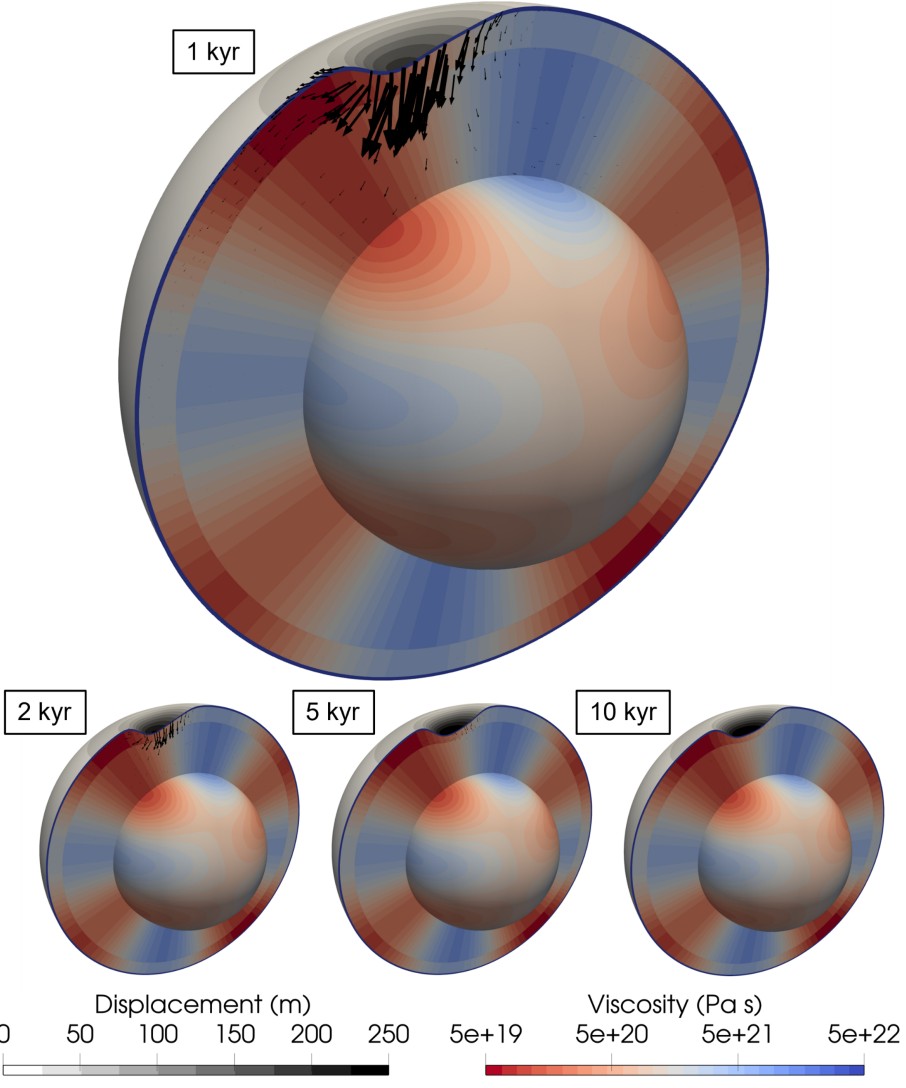

**Figure 6.** Viscoelastic deformation of a 3D spherical domain following application of a disk load at the North Pole with a compressible Burgers rheology. Snapshots are at 1 kyr, 2 kyr, 5 kyr and 10 kyr after load is applied. Red/blue = long-term, steady state viscosity (lithospheric viscosity of $\sim 10^{40}$ Pa s saturated in dark blue); greyscale = magnitude of surface displacement (surface warping exaggerated by a factor of 1500 for improved visual clarity); black arrows = velocity vectors.

470    Figure 6 shows slices through the spherical domain at times of 1 kyr, 2 kyr, 5 kyr and 10 kyr after the load is applied. Early on in the simulation, deformation velocities are large and concentrated in the upper mantle. Despite the axisymmetric nature of the load, there is a bias in their direction, with greater movement towards the neighbouring region of low viscosity. There is also a higher amplitude, shorter wavelength peripheral bulge developing on this side of the ice sheet in comparison





to the direction of the highest viscosity. After 5 kyr, the remaining deformation is now concentrated on the higher viscosity
side of the domain and has much lower velocity, consistent with expectations that the higher viscosity region has a longer
Maxwell time and therefore takes longer to reach isostatic equilibrium. By 10 kyr, the majority of deformation has finished
and the initially highly asymmetric surface displacement has gradually restored to a more symmetric pattern. At equilibrium,
the surface displacement should match the symmetry of the applied load (since we are still using radially symmetric elastic
structure).

This simulation is designed to showcase the relative ease in which G-ADOPT can be switched between different model
domain geometries. It also simultaneously demonstrates its ability to model more complex, Earth-like problems that include
compressibility, lateral viscosity variations, and transient rheologies. We believe that these aspects should render it a valuable
future tool for modelling viscoelastic surface loading problems in future work, such as those routinely encountered in GIA. In
the next section, we illustrate another key strength of G-ADOPT – namely its ability to compute automatic adjoints.

## 3.5 Adjoint twin experiments

As emphasised in the introduction, a major motivation for developing a new viscoelastic deformation code in G-ADOPT is the
availability of an automatically-derived discrete adjoint that can be used to invert any observational constraints for unknown
Earth structure and/or loading histories. Here, we use a synthetic 2-D annulus domain loaded with ice at its surface and
demonstrate this adjoint capability by performing gradient-based optimisations for ice thickness and mantle viscosity. While
this example is intentionally kept simple, our goal is to establish a clear and controlled foundation for tackling more complex
and realistic inverse problems in future work.

### 3.5.1 Synthetic forward problem

Figure 7 shows the setup of the 2D annulus used in our adjoint twin experiments, including the configuration of viscosity fields
and ice loads. In these simulations, we assume a Maxwell rheology and consider two distinct simulations that differ only in
their viscosity structure. The first adopts the same purely radial viscosity profile used in our previous benchmark cases (Table 2)
except that it uses a less stiff lithospheric viscosity of $1 \times 10^{25}$ Pa s. The second incorporates various lateral viscosity variations
(LVV) covering two orders of magnitude that are superimposed on the same radial background. These variations are defined by
Gaussian perturbations: three low-viscosity regions (i.e., $1 \times 10^{20}$ Pa s) located at equatorial parts of the core-mantle boundary
and in the upper mantle near the South Pole; and two high-viscosity features (i.e., $1 \times 10^{22}$ Pa s) in the upper mantle of the
northern hemisphere. Elastic parameters and density vary only radially and follow the values in Table 2. The mesh comprises
360 horizontal cells and 80 vertical cells, resulting in 20 cells per rheological layer.

The loading scenario consists of two ice sheets applied instantaneously at the start of the simulation via a Heaviside function.
A larger ice sheet is centred over the South Pole and has a thickness of 2 km and half-width of 20°. A second, smaller ice sheet
is offset clockwise from the north pole by 25° and has a thickness of 1 km and half-width of 10°. As in previous simulations, the
density of ice is 931 kg m$^{-3}$ and gravitational acceleration is constant, $g = 9.815$ m s$^{-2}$. Simulations are run for 10,000 years
with a 50-year timestep.





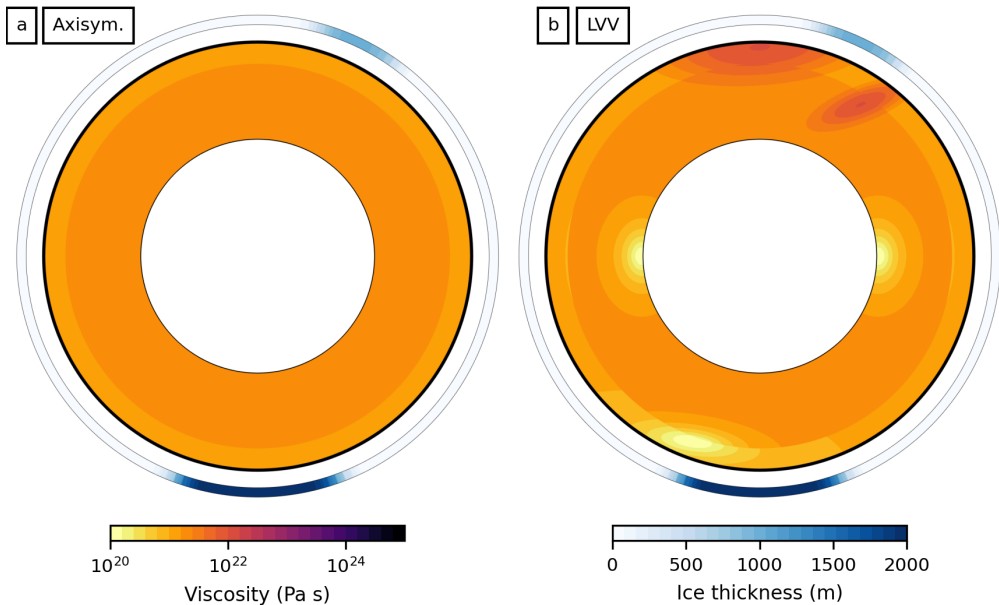

**Figure 7.** Setup of synthetic forward problem within a 2D annulus domain. (a) Axisymmetric viscosity variations. Blue external ring = thickness of the ice load instantaneously applied at start of simulation. (b) Same for the case with lateral viscosity variations (LVV).

Snapshots of the surface evolution at 3 kyr are shown in Figure 8. Looking first at the axisymmetric case, the larger South Pole ice sheet (centred at $\theta = -90°$) shows a peak downwards displacement of $\sim 320$ m, which is three times larger than the displacement under the smaller northern hemisphere ice sheet (centred at $\theta = +65°$; Figure 8a). Intervening far-field locations show fairly consistent surface uplift with an amplitude of $\sim 40$ m and ongoing rate of 3–7 mm yr$^{-1}$ (Figure 8b). At this point in the simulation, the main difference for the case with LVV occurs in the low-viscosity patch beneath the large ice sheet (between -90° and -120°). Here, we see that the shape of the subsidence is markedly more asymmetric, reaching a peak downwards displacement of -360 m and accompanied by a higher amplitude forebulge with a height of $\sim 50$ m. The rates of subsidence and uplift are also lower in comparison to the axisymmetric case, consistent with the fact that it takes less time to reach isostatic equilibrium in the lower viscosity region. In contrast, beneath the smaller ice sheet, the locally higher upper-mantle viscosity causes a delay in the surface response relative to the axisymmetric case.

Examination of the tangential displacements tells a similar story, with the biggest difference occurring above the low-viscosity zone where the surface of the axisymmetric simulation is displaced away from the ice sheet by $\sim 230$ m while that of the LVV simulation is displaced by $\sim 330$ m (Figure 8c). Interestingly, any differences are much less pronounced in the rates of tangential motion at 3 kyr (Figure 8d). Formerly examining the root-mean-squared difference in surface displacement between the two cases reveals that the radial value peaks at $\sim 0.55$ m by 2.5 kyr before gradually declining, while the tangential value peaks at just over 1.0 m and slightly later at 3.0 kyr (Figure 8e). Meanwhile the differences in surface velocity between the two





simulations are largest at the first timestep, reaching $\sim 1.0$ mm yr$^{-1}$ and $\sim 1.4$ mm yr$^{-1}$ for radial and tangential, respectively, before rapidly falling to $\sim 0.1$ mm yr$^{-1}$ by 2 kyr (Figure 8f).

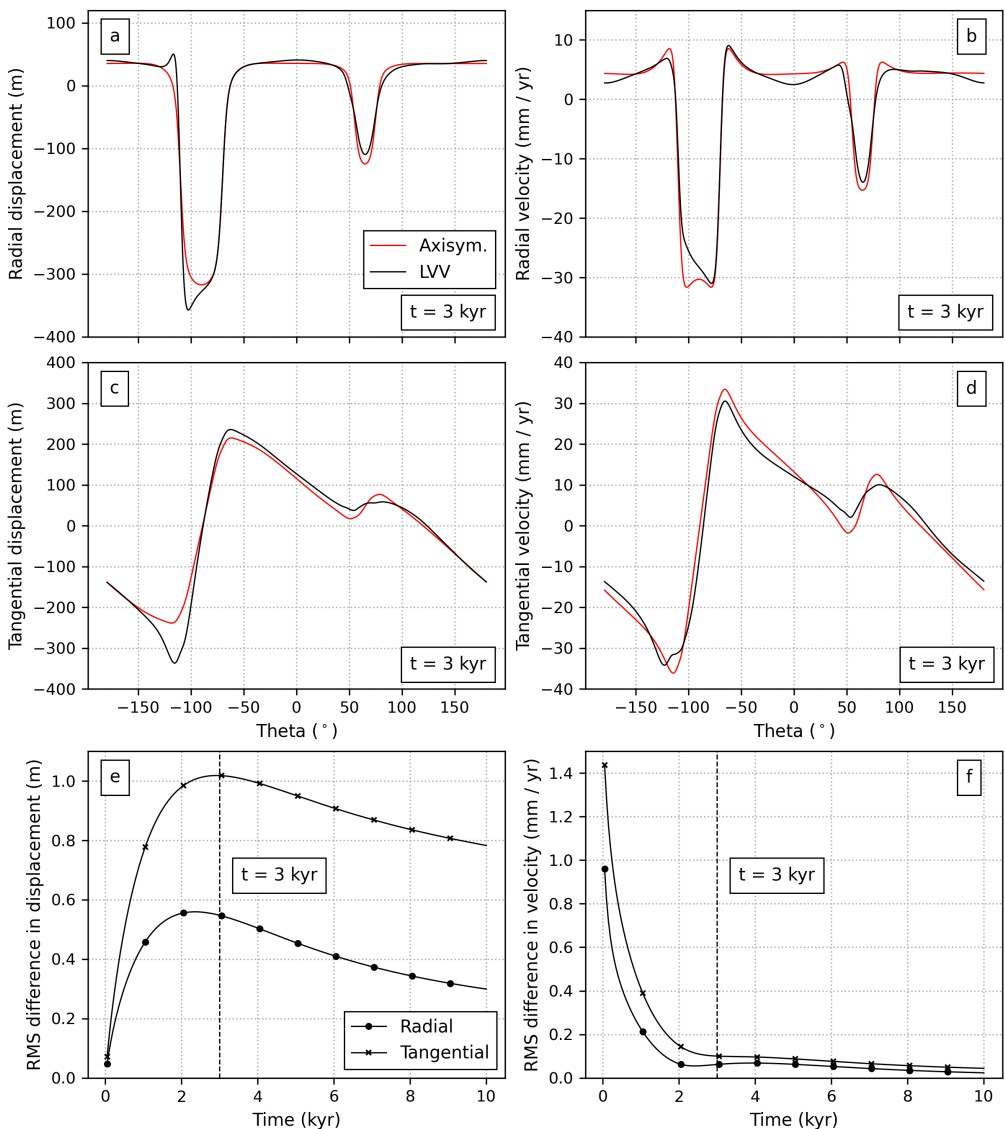

**Figure 8.** Comparison of surface displacement and surface velocity through time for 1D and 3D viscosity cases. Radial component of surface displacement (a) and surface velocity (b) at 3 kyr. (c)-(d) are corresponding tangential components. Note the large ice sheet is centred at -90° and the small ice sheet at 65°. Spatially averaged misfit between 3D and 1D viscosity case through time for surface displacement (e) and surface velocity (f).





### 3.5.2 Inversion for initial load

Armed with synthetic 'observations' of surface displacement through time for these two forward simulations, we can now demonstrate an adjoint-based optimisation. In our first demonstration, we retain knowledge of the true viscosity structure and only invert for the unknown initial ice thickness. It is important to note that this test and those that follow assume perfect observations (i.e., complete spatiotemporal data coverage with no uncertainties) and is therefore highly idealised with respect to real-world datasets.

The ice thickness that must be reconstructed is referred to as the (discrete) control vector, $\boldsymbol{c}$, and is defined using a $P1$ finite element function (i.e., linear across elements and continuous between them) on the 1D surface of the 2D annulus domain. The misfit objective function, $J(\boldsymbol{c})$, is written as

$$J(\boldsymbol{c}) = \frac{1}{N_{\Delta t} C} \sum_{n}^{N_{\Delta t}} \int_{\partial \Omega_{\text{top}}} \left( \frac{(\boldsymbol{u}^n(\boldsymbol{c}) - \boldsymbol{u}^n_{\text{obs}})^2}{u_{\text{scale}}^2} + \frac{(\boldsymbol{v}^n(\boldsymbol{c}) - \boldsymbol{v}^n_{\text{obs}})^2}{v_{\text{scale}}^2} \right) ds, \tag{39}$$

where $N_{\Delta t}$ is the number of timesteps, $C$ is the surface 'area' (i.e., circumference) of the 2D annulus, $n$ is the current timestep, $\boldsymbol{u}$ and $\boldsymbol{v}$ are current solution values for the (non-dimensional) displacement and velocity fields, respectively, $\boldsymbol{u}^n_{\text{obs}}$ and $\boldsymbol{v}^n_{\text{obs}}$ are the corresponding 'observations' from the target forward simulation, and $u_{\text{scale}} = 1 \times 10^{-4}$ and $v_{\text{scale}} = 1 \times 10^{-5}$ are factors used to scale the relative sizes of the misfit coming from displacements versus velocities. $J(\boldsymbol{c})$ is often referred to as the reduced functional, since evaluation of $J$ also requires solution of the forward model to find $\boldsymbol{u}$ and $\boldsymbol{v}$, which are in turn a function of $\boldsymbol{c}$.

Once the forward problem has been specified, G-ADOPT leverages Pyadjoint to automatically derive the adjoint problem and compute the gradient of the misfit function with respect to the control vector. To verify the accuracy of these gradients, a Taylor remainder convergence test is performed (Farrell et al., 2013; Ghelichkhan et al., 2024). It is defined as

$$|J(\boldsymbol{c} + h\,\boldsymbol{\delta c}) - J(\boldsymbol{c}) - h\nabla\boldsymbol{c} \cdot \boldsymbol{\delta c}| \longrightarrow 0 \text{ at } O(h^2), \tag{40}$$

where $\boldsymbol{c}$ is the control vector (ice thickness), $\boldsymbol{\delta c}$ is a random perturbation direction, $h \ll 1$ is a small scalar, and $J(\boldsymbol{c})$ is the reduced objective functional. The left-hand side represents the second-order Taylor remainder. As the size of the scalar is modified and if the gradient is correctly implemented, the residual should converge to zero at a rate of $O(h^2)$, which is indeed the case (Table 4).

**Table 4.** Taylor test results for the derivative of the misfit objective function with respect to ice thickness, for $h = 0.01$.

|  | $h$ | $h/2$ | $h/4$ | $h/8$ |
|---|---|---|---|---|
| Residual | $1.44 \times 10^{-5}$ | $3.61 \times 10^{-6}$ | $9.03 \times 10^{-7}$ | $2.26 \times 10^{-7}$ |
| Convergence rate | - | 2.00 | 2.00 | 2.00 |

Following the approach of Ghelichkhan et al. (2024), the computed gradient is passed to ROL and we use the Lin-Moré trust-region algorithm (Lin and Moré, 1999). This algorithm permits the placing of constraints on the control variables such as upper and lower bounds, in this case allowing enforcement of only non-negative ice thicknesses. While it would be possible



to add further regularisation terms to the objective function, for example ensuring that the ice is smooth in shape, they are not considered in this experiment. The initial guess is that there is zero ice thickness across the domain. For the 'observational data', we use the results of the synthetic forward simulation with LVV from Section 3.5.1 and invert it with both the correct LVV viscosity structure and the incorrect axisymmetric one.

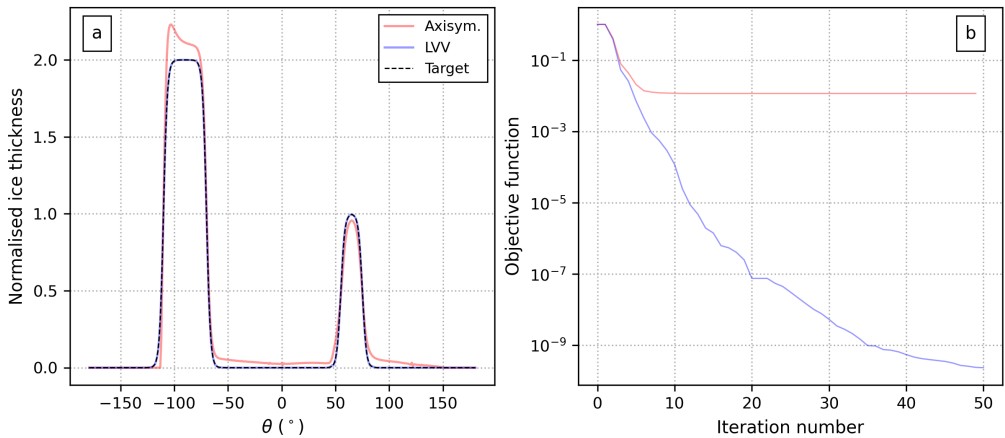

**Figure 9.** Adjoint-based optimisation for initial ice load. (a) Normalised ice thickness after 20 iterations. Black dashed line = true target load; blue = inversion using correct LVV mantle structure; red = inversion using incorrect axisymmetric mantle structure. (b) Objective (i.e., misfit) function value as a function of iteration number for both cases.

After only 20 iterations, the recovered ice thickness using the correct LVV viscosities is visually indistinguishable from the true target field and corresponds with the objective function (i.e., misfit in surface displacement and velocity) reducing by nearly seven orders of magnitude (Figure 9). In contrast, misfit for the incorrect viscosity structure initially drops by two orders of magnitude before stalling. Examining the resulting ice thicknesses shows that the recovered large ice sheet is asymmetric, with $\sim 10\%$ extra ice added to compensate for absence of the low-viscosity upper mantle region near -110°. Similarly, the smaller ice sheet is $\sim 5\%$ too thin, owing to the locally reduced viscosity in the axisymmetric model. Additionally, the optimisation introduces a thin, tapering layer of spurious ice 30–50 m-thick extending either side of the small ice sheet and into the large ice sheet (i.e., between -50° and 150°). This behaviour indicates the great value of geological constraints on ice extent, such as terminal moraines and cosmogenic exposure ages, when attempting this approach with real datasets.

Regarding the computational cost of automatic adjoint calculation, we find that Pyadjoint is very efficient. Annotation of the forward model for this 2D case takes 3% of the time required to run the forward model, while the adjoint model is only 6% slower than the forward model with taping (Table 5). This nearly matches the expected one-to-one correspondence between the cost for the forward and adjoint model for a linear forward problem (e.g. Farrell et al., 2013). We anticipate further improvements in the adjoint-to-forward-efficiency ratio for fully 3D problems, as the cost associated with solving the linear systems increases.





**Table 5.** Runtimes for the forward and adjoint models.

|  | Runtime (s) | Ratio |
|---|---|---|
| Forward model | 422 | 1.0 |
| Forward model + annotation | 434 | 1.03 |
| Forward model + annotation + adjoint model | 893 | 2.1 |

This experiment demonstrates the benefits of automatic adjoint derivation in Pyadjoint and its effective use for recovering initial loads when the viscosity structure is known. It also highlights the biases that are introduced by incorrect rheological assumptions.

### 3.5.3   Inversion for viscosity structure

We now turn to the complementary inversion problem: assuming the distribution of the initial load is known, can we recover an

unknown mantle viscosity structure? The same objective function (Eq. 39) is used, which quantifies misfit between simulated and target surface displacements and velocities at all timesteps. The target 'observational' data is again drawn from the synthetic forward simulation using viscosity structure with lateral viscosity variations and does not contain any additional noise or uncertainties.

The viscosities in this model span five orders of magnitude. In order to improve convergence rates, we found that it is helpful

to apply a logarithmic rescaling according to

$$\eta = \eta_0 10^{\eta_{\text{control}}}, \tag{41}$$

where $\eta$ is viscosity in the simulation, $\eta_0$ is a radial viscosity profile, and $\eta_{\text{control}}$ is the control field of lateral variations over which the optimisation is performed. This formulation constrains the control values to a smaller, more uniform range. $\eta_0$ is initialised using the same radial viscosity profile (although it could be any other) using a $DG0$ finite element function. $\eta_{\text{control}}$

is defined using a continuous $P1$ element and the field is initially set to zero everywhere in the domain, so that the starting guess for viscosity is equivalent to viscosities in the axisymmetric case. As before, the inversion proceeds without the need to introduce any regularisation into the objective function. A Taylor remainder convergence test confirms the accuracy of the computed gradients (Table 6).

**Table 6.** Taylor test results for the derivative of the misfit objective function with respect to viscosity.

|  | $\Delta h$ | $\Delta h/2$ | $\Delta h/4$ | $\Delta h/8$ |
|---|---|---|---|---|
| Residual | $8.50 \times 10^{-6}$ | $2.13 \times 10^{-6}$ | $5.33 \times 10^{-7}$ | $1.33 \times 10^{-7}$ |
| Convergence rate | - | 2.00 | 2.00 | 2.00 |

At iteration zero where the viscosity is still axisymmetric, the largest signals in the adjoint sensitivity kernel indicate that

viscosity should be reduced beneath the large southern ice sheet and increased beneath the smaller northern ice sheet (top-





**Figure 10.** Optimisation of viscosity structure given a known ice load. Each row shows the state of the inversion after 0, 10, 50, and 100 iterations. Left-hand column = current inference of viscosity field; central column = misfit with respect to the true viscosity structure in control-space; right-hand column = adjoint sensitivity kernel, where positive values in red indicate that decreasing viscosity reduces the misfit and negative values in blue indicate that it should be increased.





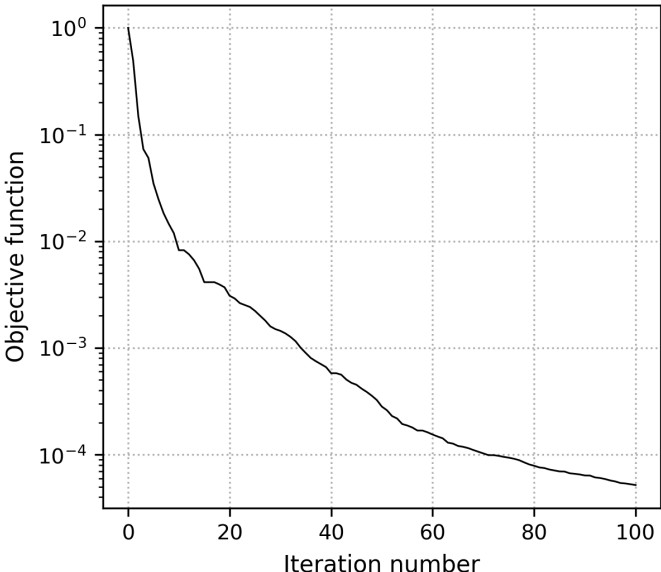

**Figure 11.** Reduction in objective (i.e., misfit) function value as a function of iteration number during viscosity inversion with known ice load.

right panel in Figure 10). This pattern is consistent with true features of the target model with lateral viscosity variations. By iteration 10, the inversion successfully recovers the bulk of the main low-viscosity zone in the upper mantle of the southern hemisphere, along with emerging high-viscosity features in the north. It is at this stage that the adjoint sensitivity kernel also begins to exhibit sensitivity to deep mantle structures on comparable levels to those in the shallow mantle. By iteration 50,

the inversion starts resolving low-viscosity regions near the core-mantle boundary, while all of the low-viscosity features are faithfully represented by iteration 100 and only some minor blurring of the high-viscosity northern hemisphere features remains (bottom-centre panel in Figure 10).

As expected, the magnitude of the adjoint sensitivity decreases with each iteration as misfit progressively reduces by over four orders of magnitude by iteration 100 (Figure 11). This behaviour reflects the strength of adjoint-based optimisation: the

algorithm first targets regions contributing most to the misfit and then gradually refines the solution in less sensitive areas. After 100 iterations, the majority of the recovered viscosity values lie within 20-25% of the target ($|\log_{10}(\eta/\eta_{\text{target}})| < 0.1$). The slightly larger errors remaining in the northern hemisphere are likely due to these being high-viscosity regions with longer Maxwell times. Here, the total amount of viscous deformation over the course of the simulations is therefore much lower and our 'observations' of surface deformation are less sensitive to details of the structure in this part of the domain.

**3.5.4  Simultaneous inversion for viscosity and load**

The previous two experiments demonstrate G-ADOPT's capability to recover either the ice load or the lateral viscosity structure under the assumption that the other is known. The natural follow-up question is whether it is possible to jointly invert for both





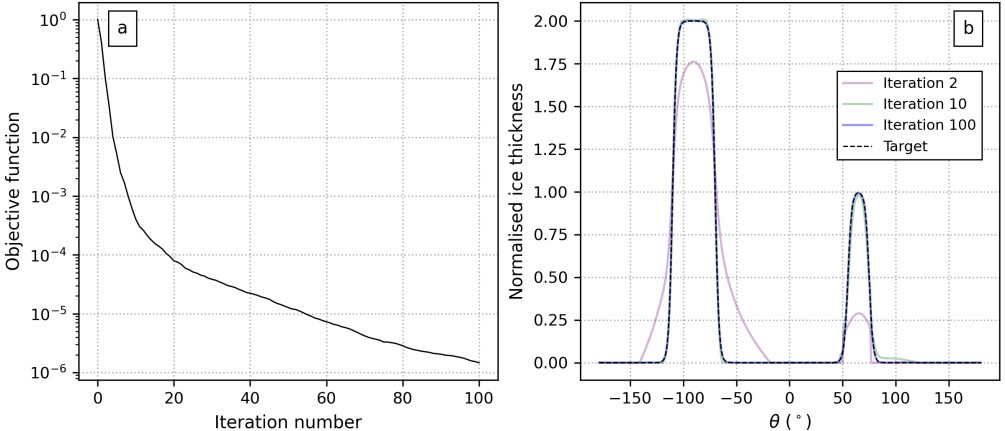

**Figure 12.** Adjoint-based joint optimisation for both initial ice load and viscosity structure. (a) Objective (i.e., misfit) function value as a function of iteration number. (b) Normalised ice thickness. Black dashed line = true target load; pink = inversion after 2 iterations; green = 10 iterations; blue = 100 iterations.

load and viscosity structure when neither is known? To investigate this question, we use a setup that mirrors that of the previous two cases but with a control vector now containing both fields. We initialise the inversion with zero ice thickness and the

axisymmetric viscosity structure. Again, we assume no observational uncertainty and have not implemented any regularisation in the objective function.

Despite this inversion being ostensibly more challenging than the previous two experiments, the results are highly encouraging. Over the course of 100 iterations, the objective function decreases by nearly six orders of magnitude (Figure 12a). Already by iteration 10, optimised ice thicknesses show strong similarities with the true target distribution (Figure 12b) and key features

of the viscosity structure are visible including the low-viscosity region in the southern hemisphere and emerging high-viscosity features in the north (Figure 13c). By iteration 100, the ice load is visually indistinguishable from the target and the viscosity field is as similar to the true structure as in the viscosity-only inversion from Section 3.5.3.

A critical factor in the success of this joint inversion is finding an appropriate balance between the contribution to the gradient from the two control fields (i.e., ice and viscosity). We have used the $L_2$ derivative for optimisation in all experiments,

as opposed to the more traditional Euclidean $l_2$ derivative that is defined with respect to the control vector's discrete degrees of freedom. The $L_2$ formulation is more appropriate because it properly accounts for variations in mesh cell size and, more importantly, for the fact that the ice is defined on a surface mesh that has different geometric dimensions to the volumetric viscosity field. Specifically, the optimisation is driven by the gradients $\left[\frac{\partial J}{\partial h_{\text{load}}}\right]_{L_2(\partial \Omega_{\text{top}})}$ and $\left[\frac{\partial J}{\partial \eta}\right]_{L_2(\Omega)}$ such that sensitivity of the objective function to perturbations $\delta h$ and $\delta \eta$ in thickness and viscosity is given by

$$\delta J = \int_{\partial \Omega_{\text{top}}} \left[\frac{\partial J}{\partial h_{\text{load}}}\right]_{L_2(\partial \Omega_{\text{top}})} \delta h_{\text{load}} \, ds + \int_{\Omega} \left[\frac{\partial J}{\partial \eta}\right]_{L_2(\Omega)} \delta \eta \, dx. \tag{42}$$





**Figure 13.** Adjoint-based joint optimisation for both initial ice load and viscosity structure. (a) True target viscosity field and surface ice load. (b) Inversion results after 2 iterations. (c) Same after 10 iterations. (d) Same after 100 iterations.

We found that for the Euclidean $l_2$ approach that uses a simple summation of partial derivatives with respect to the discrete degrees of freedom, the inversion became dominated by sensitivity of misfit to ice thickness, leading to suboptimal results. In such cases, the inversion typically stalled, yielding a noisy ice-thickness field and failing to recover the deep mantle or northern hemisphere viscosity variations (results not shown). Importantly, the integral $L_2$-approach is delivered automatically
in G-ADOPT through combined use of Firedrake, pyadjoint and ROL.



In summary, using a suite of idealised experiments, we have demonstrated that G-ADOPT has the ability to automatically and successfully perform adjoint-based inversions for both initial ice load and/or lateral variations in mantle viscosity. This capability is powerful since it circumvents the need for lengthy derivations of the appropriate adjoint formulations for each viscoelastic loading problem.

## 4   Discussion, limitations and future work

A wide range of problems in geodynamics require modelling the viscoelastic response of the solid Earth to evolving surface loads. Our long-term objective is to develop a generalised framework in G-ADOPT capable of tackling this broad class of problems. Nevertheless, our more immediate motivation is to develop more advanced GIA models that incorporate lateral Earth structure, potentially complex rheologies, and use data assimilation techniques to constrain uncertain parameters such as ice histories and Earth structure. In this context, several aspects of the current implementation – its capabilities, limitations and future directions – warrant discussion.

We have demonstrated the potential of G-ADOPT to represent Earth-like viscoelastic behaviour under surface loading. The framework already supports spherical geometries, elastic compressibility, large lateral variations in viscosity, and complex rheologies, including transience via the internal variable formulation (Al-Attar and Tromp, 2014; Crawford et al., 2017). Its development is timely, as recent studies have argued that transient deformation is required to reproduce GIA observations (Simon et al., 2022; Paxman et al., 2023; Lau, 2024). In G-ADOPT, more sophisticated transient models, such as extended Burgers rheologies, can be implemented in a straightforward manner by introducing additional internal variables. This capability provides a unified setting in which competing rheological scenarios can be systematically explored.

The most important physical process still absent is self-gravity, which requires solving Poisson's equation for the gravitational potential subject to boundary conditions at infinity. Implementing these conditions in discretisation-based methods is challenging. Approaches explored in previous studies include: (i) extending the computational domain with an outer mesh to approximate infinity (e.g. May and Knepley, 2011); (ii) applying a Dirichlet-to-Neumann boundary condition, whereby the analytic spectral solution outside of the domain is imposed as a weak boundary term (e.g., Chaljub and Valette, 2004); and (iii) employing semi-infinite elements with basis functions specifically chosen to reproduce the appropriate asymptotic decay (e.g., Gharti and Tromp, 2017). An alternative strategy is to couple the numerical model to an external analytic solution for gravity (Zhong et al., 2022). Our future work will focus on a hybrid of the first two approaches.

Once self-gravity is included, extending the framework to capture rotational feedbacks of load redistribution and deformation (i.e. true polar wander; Mitrovica et al., 2005) will be relatively straightforward. Together these additions will complete the physics of the sea-level equation, which links ice-sheet evolution to global sea-level change (Farrell and Clark, 1976; Kendall et al., 2005). A practical component of this equation is the continent-ocean function, which defines the ocean load. Since shoreline migration must be represented dynamically, we will adopt approaches compatible with automatic differentiation – for example, locally smoothed (e.g., sigmoidal) approximations or a level-set method that is already implemented in G-ADOPT.



Another key strength of G-ADOPT is its suitability for extension to large-scale GIA problems requiring high spatiotemporal resolution and/or long durations using high-performance computers. Scaling tests on the Gadi supercomputer show excellent parallel performance to at least ~6000 cores. Unlike explicit approaches (e.g. Lloyd et al., 2024), implicit methods are not limited by the Maxwell relaxation time, allowing timesteps long enough to capture full glacial cycles, even for viscosity contrasts spanning several orders of magnitude, without incurring excessive computational cost. Moreover, G-ADOPT maintains robust accuracy across a wide range of mesh resolutions and timestep sizes, underscoring its reliability in both regional and global applications. These properties naturally position the framework to scale to continental- and global-scale adjoint inversions, where computational efficiency and flexibility in rheological assumptions will be essential.

While adjoint methods are well established in mantle convection modelling (e.g., Bunge et al., 2003; Li et al., 2017; Ghelichkhan et al., 2021), their use in GIA is more recent. Initial studies have already demonstrated their diagnostic power: adjoint sensitivity kernels clearly expose the limitations of 1D Earth assumptions and the strong influence of lateral viscosity variations on GIA observables (e.g., Al-Attar and Tromp, 2014; Crawford et al., 2018; Kim et al., 2022; Lloyd et al., 2024). G-ADOPT provides a fully automated, scalable, and extensible inversion framework that eliminates the need for manual adjoint derivations or simplifying assumptions to manage non-linearity and coupling. This functionality is achieved by combining: (i) Firedrake, which separates model formulation from implementation within a composable finite element environment (Rathgeber et al., 2016); (ii) Pyadjoint, which derives discrete adjoints directly from variational problems expressed in unified form language (Farrell et al., 2013; Mitusch et al., 2019); and (iii) ROL, which supplies high-performance optimisation algorithms with checkpointing and optional parameter constraints. Together, these components support a streamlined workflow for forward simulation, adjoint construction, and gradient-based inversion.

It is important to acknowledge that the inverse experiments presented here commit the classic "inverse crime": the synthetic observations used for optimisation were generated by the same numerical model used for inversion and were assumed to be noise free and complete in space and time. This deliberate choice allowed us to isolate the framework's performance under idealised conditions and confirm its correctness. The next step is to apply G-ADOPT with real observational constraints and will require formal uncertainty quantification, careful treatment of data–model misfit, and potentially hybrid assimilation strategies that combine multiple observation types. As misfit functions become more complex – for example, by incorporating discrete relative sea-level indicators or palaeo-ice extent constraints – regularisation strategies such as smoothing, damping and prior-based parameter constraints will be essential to ensure stable adjoint solutions and interpretable model updates.

In summary, G-ADOPT delivers a flexible, extensible, and computationally scalable framework for adjoint-enabled modelling of viscoelastic surface-loading problems, with a particular focus on GIA. With the planned inclusion of self-gravity, rotational feedbacks, and dynamic shoreline migration, it will provide a next-generation, data-assimilating system that can directly link palaeo and modern sea-level observations to the coupled evolution of ice sheets and Earth's interior.





## 5 Conclusions

In this study, we have extended G-ADOPT to model viscoelastic deformation of the solid Earth under evolving surface loads. We have demonstrated its ability to run simulations in 2D and 3D domains, in Cartesian and spherical geometries, and incorporating effects such as elastic compressibility, lateral viscosity variations, and non-Maxwell rheologies (including transient behaviour). Where possible, results were benchmarked against analytical solutions and published community models. We further showed that G-ADOPT scales efficiently on high-performance computing systems, enabling simulations with high spatial

resolution and long time spans. A key outstanding challenge is the incorporation of self-gravity, which will be the focus of ongoing development.

Our work also lays the foundation for addressing several long-standing challenges in GIA research. As observational constraints become more diverse and models of mantle rheology more complex, there is an increasing need for inversion frameworks that can flexibly integrate these data while maintaining physical and numerical rigour. A major strength of G-ADOPT is

its automatic derivation of adjoint sensitivity kernels, which enables inverse optimisation without the need for manual derivation of adjoint equations. Using synthetic twin experiments, we demonstrated simultaneous recovery of both ice-loading history and mantle structure, showing that G-ADOPT's adjoint implementation is robust, accurate, and suitable for application to real geophysical inverse problems. At the same time, our results highlight the inherent biases that arise when inverting for ice histories with inaccurate viscosity structures.

G-ADOPT has been developed in alignment with FAIR (Findable, Accessible, Interoperable, Reusable) software principles (Wilkinson et al., 2016). We aim to lower barriers to adoption, facilitate community benchmarking, and provide a robust foundation for collaborative development. Looking forward, the composable design of G-ADOPT opens the way to incorporating new classes of observational constraints into solid-Earth deformation problems. The integration of Firedrake, Pyadjoint, and ROL enables these extensions with minimal disruption to existing workflows.

Several physical processes central to sea-level modelling remain to be incorporated including self-gravitation, rotational feedbacks, shoreline migration, and ocean loading. While these aspects are non-trivial to implement, the framework's modular design ensures that they can be added incrementally. Embedding G-ADOPT within the Firedrake ecosystem also enables coupling with other Earth system components, such as the Icepack ice-sheet model (Shapero et al., 2021), raising the possibility of coupled Earth–climate simulations capable of data assimilation over full glacial cycles.

*Code and data availability.* For the specific components of G-ADOPT, including the full scripts of the simulations used in this paper see https://doi.org/10.5281/zenodo.16925270 (Scott, 2025). For the specific components of Firedrake project used in this paper see https://zenodo.org/records/16795889 (Mitchell et al., 2025). For ongoing developments of the G-ADOPT code base see https://github.com/g-adopt/g-adopt

*Author contributions.* All authors had significant input on the design, development and validation of this study. All authors contributed towards writing the manuscript.



*Disclaimer.* Publisher's note: Copernicus Publications remains neutral with regard to jurisdictional claims in published maps and institutional affiliations.

*Financial support.* This research was supported by the Australian Research Council Special Research Initiative, Australian Centre for Excellence in Antarctic Science (Project Number SR200100008). This research has been supported by AuScope, under the CoastRI NCRIS program, the Australian Research Data Commons (ARDC) under G-ADOPT platform grant PL031, and Geoscience Australia. It was also

supported by the Australian Research Council under grants DE220101519, DP170100058 and DP220100173. This research was undertaken with the assistance of resources from the *National Computational Infrastructure* (NCI Australia), a National Collaborative Research Infrastructure Strategy (NCRIS) enabled capability supported by the Australian Government.

*Acknowledgements.* Numerical simulations were undertaken on the Gadi supercomputer at the National Computational Infrastructure (NCI) in Canberra, Australia, which is supported by the Australian Commonwealth Government. The authors are grateful to the entire Firedrake,

Dolfin-Adjoint and ROL development teams for support and advice at various points of this research.



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
