# Peer review of "Automated forward and adjoint modelling of viscoelastic deformation of the solid Earth"

_EGUsphere, 2025_

## Author Comment (AC1)

**Rebuttal for egusphere-2025-4168: Automated forward and adjoint modelling of viscoelastic deformation of the solid Earth**

William Scott[1], Mark Hoggard[1], Thomas Duvernay[1], Sia Ghelichkhan[1, 2], Angus Gibson[1],
Dale Roberts[1], Stephan C. Kramer[3], and D. Rhodri Davies[1]

[1]Research School of Earth Sciences, The Australian National University, Canberra, ACT, Australia.
[2]Institute for Water Futures, The Australian National University, Canberra, ACT, Australia.
[3]Department of Earth Science and Engineering, Imperial College London, London, UK.

**Correspondence:** William Scott (william.scott1@anu.edu.au)

We thank the two reviewers for their time and considered feedback on our manuscript. Both of them state that our study is a valuable contribution to GIA modelling that lays a pathway to addressing new questions in the field. Between them, they raise four helpful discussion points that we outline and respond to in turn below. In addition, on the basis of verbal community feedback, we have also made a minor edit to the formulation of our governing equations and we outline the associated changes at the end of this rebuttal.

*Reviewer 1 asks: "Line 586: Regularisation is noted as an important mechanism for stabilizing the adjoint solution (see also line 688), particularly when observations are indirect outputs rather than primary solution fields. In the LVV inversion, the gradient appears significantly steeper across the upper–lower mantle viscosity transition due to the sharp contrast. Would introducing spatial or parameter-space regularisation help improve the reconstruction of features that span this boundary? If so, could the authors comment on how such regularisation might be formulated within the objective function without excessively smoothing meaningful structures?"*

We agree that the introduction of regularisation terms that smooth the underlying viscosity field has the potential to reduce how well any sharp viscosity contrasts will be resolved and that implementing specific regularisation strategies will help with resolving features at these depths. To that end, in the LVV inversions presented herein, we implement a form of parameter-space regularisation (Equation 41 of our original submission) by making the control a continuous viscosity variation field on top of a fixed background radial profile that contains discontinuous jumps, which encourages the phase-boundary viscosity increase to be preserved during the inversion. In reality, this approach necessitates prior knowledge on the locations of any such sharp transitions. Furthermore, the physics of GIA itself apply a filtering effect on this structure, such that sharp viscosity contrasts away from the shallow mantle start to be indistinguishable from smoother transitions from the perspective of surface observables. This phenomenon is behind the apparent switch in polarity of the adjoint kernels across the upper-lower mantle transition and general smoothing of short-wavelength features into longer ones (even though there is no formal regularisation in these inversions). We do not yet have a good enough understanding for what strategies will be most effective when working on real Earth data and so, whilst

acknowledging that this will require careful consideration in future studies, we have avoided discussing it in too much detail at this point.

*Reviewer 1 asks: "Line 685: It would be helpful to clarify how observational uncertainty is intended to be handled. Will observational variance be incorporated into the cost function weighting during inversion, or is the aim instead to estimate posterior uncertainty bounds on the inferred parameters—or both? Clarifying this point would help readers understand the intended interpretational framework of the inversions."*

We agree that we needed to clarify how observational uncertainties would be handled. The Reviewer is correct – our intention is to include these uncertainties as weights within the cost function. While the computational cost of our simulations renders the use of probabilistic or ensemble methods to map this uncertainty into inferred model parameters challenging, we are hoping that second-order adjoint methods quantifying uncertainties using Hessian-based inference might provide a means to more efficiently tackle this propagation problem (e.g. Yu et al., 2025). We have edited this paragraph of our revised discussion to reflect these two points.

*Reviewer 2 asks: "My suspicion is that down the line many key problems of interest may require resolutions that necessitate scaling beyond 6,000 cores. What changes to the numerical framework would be required to scale to 10,000s of CPUs? A switch from an algebraic to geometric (matrix-free) multigrid preconditioner? The paper would benefit from briefly describing how this could potentially be achieved within the G-ADOPT framework."*

We agree that even higher resolution simulations than we have run here are likely to be necessary for tackling some of the major outstanding problems in GIA. In our code development so far, we have run a small number of tests using cores numbering in the tens of thousands. While the results of our current implementation remained satisfactory, we did notice a comparative increase in the fraction of the computation time that was spent assembling the preconditioner. In its current iteration, our code makes use of an algebraic multigrid preconditioner, which has the advantage of being suitable for both structured and unstructured meshes. As the Reviewer has correctly pointed out, switching to a geometric multigrid preconditioner could save on this computational cost, although with the caveat that it would only be suitable for use with structured meshes. This paragraph has been revised in our resubmission to reflect this discussion.

*Reviewer 2 asks: "While the numerical framework is capable of simulating models with large viscosity variations, the underlying constitutive model is for a linear viscoelastic material. What possibility is there to also integrate nonlinear constitutive models, such as the viscoelastic-plastic models commonly used in the tectonic geodynamics community and in Shijie Zhong's codes for modeling loading induced deformation (e.g., Zhong and Watts, 2013, https://doi.org/10.1002/2013JB010408)? There is a note on line 165 that the constitutive formulation can be extended to non-linear constitutive equations, but the manuscript would benefit from more detail on exactly what can be reasonably achieved within the current framework within the context of other relevant studies."*

We agree that it is important to demonstrate the ability of our code to use non-linear constitutive models. We have therefore now added this functionality into the code base. In the manuscript, we present an example case using a

composite power-law rheology from Kang et al. (2022). Using the same 1D loading scenarios as in the benchmarks of Weerdesteijn et al. (2023), we demonstrate and compare its impact in comparison to the Maxwell and Burgers rheologies (see revised Section 3.3 and Figure 5). We thank the Reviewer for their encouragement to undertake this
60 important addition.

*Verbal community feedback on symmetry in our governing equations:*

In addition to these four comments, we also received helpful feedback concerning the formulation of our weak form of the governing equations (original manuscript Equation 32). It was pointed out that these equations were not symmetric in form due to the presence of jump terms introduced in Equations (28) and (29). Although this formulation is not
65 incorrect, using a symmetric formulation permits a wider choice of numerical methods for solving them, potentially allowing faster solves. We have therefore recast them in their symmetric form (revised Equation 32). In addition to this advantage, it also makes them more faithful to the underlying assumption of hydrostatic equilibrium in the background state. We have made the associated minor modifications necessary to our code base and confirmed that all numerical results are consistent with the previous formulation.

**70 References**

Kang, K., Zhong, S., Geruo, A., and Mao, W.: The effects of non-Newtonian rheology in the upper mantle on relative sea level change and geodetic observables induced by glacial isostatic adjustment process, Geophysical Journal International, 228, 1887–1906, 2022.

Weerdesteijn, M. F., Naliboff, J. B., Conrad, C. P., Reusen, J. M., Steffen, R., Heister, T., and Zhang, J.: Modeling viscoelastic solid Earth deformation due to ice age and contemporary glacial mass changes in ASPECT, Geochemistry, Geophysics, Geosystems, 24, 75   e2022GC010 813, 2023.

Yu, Z., Al-Attar, D., Syvret, F., and Lloyd, A. J.: Application of first-and second-order adjoint methods to glacial isostatic adjustment incorporating rotational feedbacks, Geophysical Journal International, 240, 329–348, 2025.